# Two distinct subpopulations of human stem-like memory T cells exhibit complementary roles in self-renewal and clonal longevity

Danai Koftori[1‡], Charandeep Kaur[1‡], Laura Mora Bitria[1‡], Yan Zhang[2], Linda Hadcocks[2], Ada W. C. Yan[1], Piotr F. Burzyński[1], Kristin Ladell[3], Daniel E. Speiser[4], Katrina M. Pollock[1,5], Derek Macallan[2], Becca Asquith[1]*

1 Department of Infectious Disease, Imperial College London, London, United Kingdom, 2 Institute for Infection and Immunity, St George's, University of London, London, United Kingdom, 3 Division of Infection and Immunity, Cardiff University School of Medicine, Heath Park, Cardiff, United Kingdom, 4 Department of Oncology, Lausanne University Hospital and University of Lausanne, Lausanne, Switzerland, 5 Department of Paediatrics, University of Oxford, Oxford, United Kingdom

‡ DK, CK, and LMB share first authorship on this work.
* b.asquith@imperial.ac.uk

## Abstract

T stem cell-like memory cells ($T_{SCM}$ cells) are considered to be essential for the maintenance of immune memory. The $T_{SCM}$ population has been shown to have the key properties of a stem cell population: multipotency, self-renewal and clonal longevity. Here we show that no single population has all these stem cell properties, instead the properties are distributed. We show that the human $T_{SCM}$ population consists of two distinct cell subpopulations which can be distinguished by the level of their CD95 expression (CD95int and CD95hi). Crucially, using long-term *in vivo* labelling of human volunteers, we establish that these are distinct populations rather than transient states of the same population. These two subpopulations have different functional profiles *ex vivo*, different transcriptional patterns, and different tissue distributions. They also have significantly different TREC content indicating different division histories and we find that the frequency of CD95hi $T_{SCM}$ increases with age. Most importantly, CD95hi and CD95int $T_{SCM}$ cells also have very different dynamics *in vivo* with CD95hi cells showing considerably higher proliferation but significantly reduced clonal longevity compared with CD95int $T_{SCM}$. While both $T_{SCM}$ subpopulations exhibit considerable multipotency, no single population of $T_{SCM}$ cells has both the properties of self-renewal and clonal longevity. Instead, the "stemness" of the $T_{SCM}$ population is generated by the complementary dynamic properties of the two subpopulations: CD95int $T_{SCM}$ which have the property of clonal longevity and CD95hi $T_{SCM}$ which have the properties of expansion and self-renewal. We suggest that together, these two populations function as a stem cell population.

**Data availability statement:** All relevant data are within the paper and its Supporting Information files.

**Funding:** BA is funded by the Wellcome Trust (103865Z/14/Z, 220794/Z/20/Z), the European Union Seventh Framework Programme (FP7/2007–2013) under grant agreement 317040 (QuanTI), the European Union H2020 programme under grant agreement 764698 (QUANTII) and Leukemia and Lymphoma Research (15012). AWCY was supported by an Imperial College Research Fellowship. KMP was supported by a St Mary's Development Trust fellowship. The EAVI programme was funded by the European Union's Horizon 2020 research and innovation programme grant 681137. DCM was partially funded by the Jefferiss Trust. CK and LMB received salaries from the European Union H2020 programme under grant agreement 764698 (QUANTII). Infrastructure support was provided by the NIHR Imperial Biomedical Research Centre and the NIHR Imperial Clinical Research Facility. The funders did not play any role in the study design, data collection and analysis, decision to publish, or preparation of the manuscript.

**Competing interests:** The authors have declared that no competing interests exist.

**Abbreviations:** CT, cycle threshold; CTVcell trace violetMFI, median fluorescence intensity; IQR, interquartile range; $T_{CM}$, T central memory; $T_{EM}$T effector memory$T_N$, T naïve; TEMRAT effector memory expressing RATREC, T cell receptor excision circle; $T_{SCM}$, T stem-cell memory.

## Introduction

T cell memory is central to protective immunity yet how memory is maintained is poorly understood. In 2011, in a seminal paper, Gattinoni and colleagues identified, in humans, a population of antigen-experienced T cells with stem cell-like properties that included a high capacity for self-renewal *in vitro* [1]. It was observed that these T stem cell-like memory cells ($T_{SCM}$ cells) were long-lived and multipotent, i.e., they could differentiate into all other T cell memory subsets including central memory ($T_{CM}$), effector memory ($T_{EM}$) and effector memory expressing RA ($T_{EMRA}$) [1,2]. It was suggested that $T_{SCM}$ cells are a dedicated population of cells whose role is to maintain T cell memory; i.e., they serve as a long-lived reservoir that can maintain themselves for years and be called upon to differentiate into effector populations in the event of re-exposure to cognate antigen. Since then, there has been considerable interest in $T_{SCM}$ cells as preventive and therapeutic tools and in the role of $T_{SCM}$ in the longevity of the vaccine-induced T cell response [3,4].

$T_{SCM}$ cells are not the only candidate for the stem cell-like precursor population. Others have argued that $T_{CM}$ rather than $T_{SCM}$ are the main population responsible for maintaining memory [5]. This position is particularly supported in murine models. Although $T_{SCM}$ cells were first described in mouse models of graft v host disease [6], can be generated with murine cells *in vitro* and are detectable in mice *in vivo* [7], it has proved difficult to identify their antigen-specificity or to generate them following *in vivo* challenge. Furthermore, cell fate–tracking experiments in mice (using genetic barcoding and single-cell transfer), showed that $T_{CM}$ cells have the capacity to self-renew and that a single murine $T_{CM}$ cell could reconstitute immune protection against an otherwise lethal pathogen [8,9].

Outside of murine models, $T_{SCM}$ cells are less controversial and evidence for the stem-cell like behavior of $T_{SCM}$ cells has continued to accumulate. $T_{SCM}$ cells were identified in non-human primates (rhesus macaques and pigtail macaques) and importantly these $T_{SCM}$ populations could be readily generated following acute infection [10,11]. In humans, following hematopoietic stem cell transfer, the number of persisting donor cells was strongly correlated with the number of $T_{SCM}$ cells infused. Furthermore, stable isotope labelling studies of human volunteers established that, even in healthy, lymphocyte-replete individuals, $T_{SCM}$ cells were long-lived and had the dynamic properties of self-renewal and clonal longevity consistent with the hypothesis that $T_{SCM}$ cells are stem-like cells responsible for maintaining immune memory in humans [12,13].

The stable isotope labelling studies also made an unexpected finding: instead of being a single homogenous population as previously assumed, there was strong evidence that both CD4$^+$ and CD8$^+$ $T_{SCM}$ populations were in fact heterogeneous and consisted of subpopulations with different dynamics [12]. Stable isotope labelling is a destructive assay in that once cells are sorted, DNA is isolated, and the identity of the cell is lost. Therefore, it was not possible to go back and obtain the phenotype of the subpopulations with differential kinetics. Consequently, although there was strong indirect evidence for heterogeneity the subpopulations could not be identified.

Here, we identify these $T_{SCM}$ subpopulations in CD8$^+$ T cells. We first use Ki67 expression to identify cell subpopulations with different proliferation rates and establish that CD95 expression distinguishes between the subpopulations. We show that $T_{SCM}$ cells with high levels of CD95 expression (henceforth CD95hi) have significantly higher levels of Ki67 expression than $T_{SCM}$ cells with intermediate levels of CD95 expression (henceforth CD95int). We then investigate the functional properties of the CD95hi and CD95int $T_{SCM}$ subpopulations. We show that CD95hi $T_{SCM}$ cells have significantly higher self-renewal and effector functionality compared to CD95int $T_{SCM}$ cells. Transcriptionally, we show that CD95hi $T_{SCM}$ cells are closer to central memory cells while CD95int cells are closer to naïve T ($T_N$) cells. We also show that CD95int and CD95hi $T_{SCM}$ cells have different physical locations with CD95int $T_{SCM}$ enriched in lymph nodes while CD95hi $T_{SCM}$ are enriched in the blood. Returning to their dynamics, we show that CD95hi $T_{SCM}$ cells have a significantly lower TREC content than CD95int $T_{SCM}$ cells and that their frequency increases significantly with donor age. Finally, we go back *in vivo* and use stable isotope labelling, this time sorting on CD95 expression, and show that CD95hi and CD95int $T_{SCM}$ populations have different dynamics *in vivo*. Importantly, the stable isotope traces for the two populations remain distinct for the duration of the labelling experiment (140 days), ruling out the possibility that there is rapid interconversion between the two subpopulations and establishing them as bona fide distinct subsets. We fitted different models to investigate the lineage relationship between the cell subpopulations. We find that, of the models we consider, a linear model in which upon meeting cognate antigen, $T_N$ undergo clonal expansion and differentiate into CD95int $T_{SCM}$ which in turn differentiate into CD95hi $T_{SCM}$, is most consistent with the data. Furthermore, we find that the CD95int subpopulation, but not the CD95hi subpopulation, harbors long-lived clones, with half lives of more than 10 yrs. This finding is independently verified by the quantification of tetramer-positive cells post-yellow fever vaccination.

While we identify many differences between the CD95int and CD95hi $T_{SCM}$ subpopulations, it is the difference in their dynamics that is critical. CD95int $T_{SCM}$ have the longest life span of any memory subpopulation identified to date [14]. This long life span will be central to their ability to maintain immune memory.

## Results

### CD95 expression identifies $T_{SCM}$ subpopulations with differential *ex vivo* proliferation

Our aim was to identify subpopulations of CD8$^+$ $T_{SCM}$ cells (CD8$^+$CD3$^+$CD45RA$^+$CCR7$^+$CD95$^+$) with different kinetics. We therefore stained human peripheral blood mononuclear cells (PBMC) with a panel of markers ("Materials and methods") and, by backgating on the Ki67 positive cells established that the median fluorescence intensity (MFI) of CD95 expression was significantly higher in Ki67$^+$ $T_{SCM}$ cells than in Ki67$^-$ $T_{SCM}$ cells ($P = 3 \times 10^{-7}$, paired Wilcoxon, $N = 25$; Fig 1A). We therefore classified $T_{SCM}$ cells as having either intermediate or high CD95 expression (henceforth CD95int and CD95hi, respectively) with the separation between the populations based on backgating of Ki67 with gates set to provide maximum discrimination of Ki67-positive and Ki67-negative cells ("Materials and methods"). Using this definition, we showed that CD95hi $T_{SCM}$ cells had significantly higher levels of Ki67 expression than both CD95int $T_{SCM}$ cells and $T_{CM}$ (CD8$^+$CD3$^+$CD45RA$^-$CCR7$^+$) cells (Fig 1B, $P = 0.003$, $P = 0.002$, respectively; paired Wilcoxon test two tailed). To validate this finding, we investigated an independent dataset and again the same pattern was observed with CD95hi $T_{SCM}$ cells containing a significantly higher proportion of Ki67-expressing cells than both CD95int $T_{SCM}$ and $T_{CM}$ (Fig A in S1 Text).

### CD95hi $T_{SCM}$ cells exhibit increased self-renewal and functionality compared to CD95int $T_{SCM}$ cells

Self-renewal, i.e., the ability to maintain phenotype upon division, is a crucial property of stem cells. We investigated the self-renewal properties of CD95int and CD95hi $T_{SCM}$ cells together with $T_{CM}$ cells *ex vivo*. CD8$^+$ CD95int, CD95hi and $T_{CM}$ cells were isolated by FACS from PBMC from healthy donors (see Fig B in S1 Text for representative gating). Cells were cultured for 7 days with homeostatic cytokines (IL-7/IL-15, IL-15 or IL-2) [15–17] and their self-renewal, defined as the percentage of divided cells that had retained their input phenotype, was quantified where divided cells were identified by the dilution of cell trace violet. The self-renewal of CD95hi $T_{SCM}$ was consistently higher than that of CD95int $T_{SCM}$ or $T_{CM}$ in

PLOS Biology

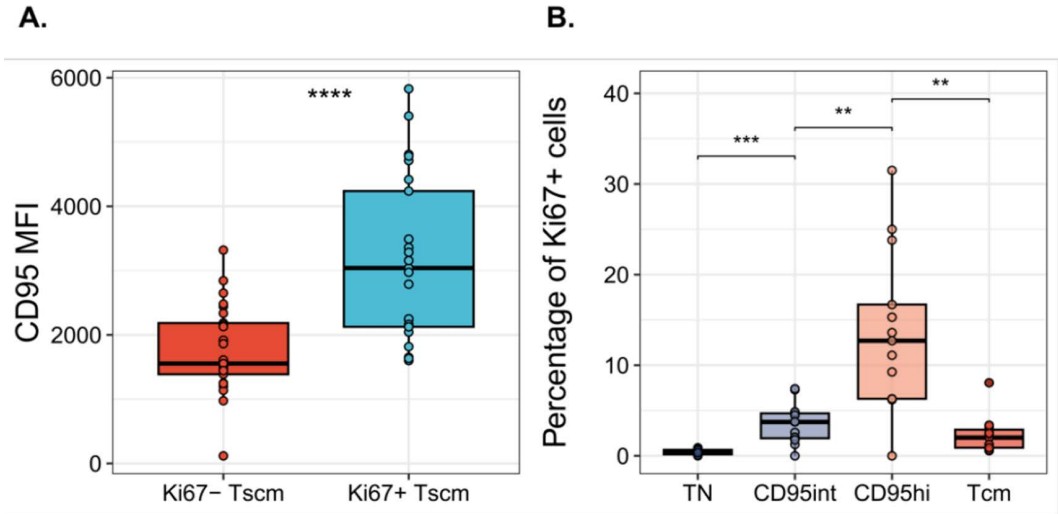

**Fig 1. Ki67+ T$_{SCM}$ cells exhibit increased CD95 expression.** T cells from healthy volunteers were isolated and stained with Ki67 and canonical T cell differentiation markers. **(A)** Median Fluorescence Intensity (MFI) of CD95 for Ki67$^-$ and Ki67$^+$ T$_{SCM}$ cells. Each dot represents one donor ($N$ = 25, Cohort 1); $P=3 \times 10^{-7}$ paired Wilcoxon **(B)** T$_{SCM}$ cells are classified as CD95hi and CD95int and the percentage of Ki67$^+$ cells is compared among T$_N$, CD95int, CD95hi and T$_{CM}$ subsets. Each dot represents one donor ($N$ = 15, from Cohort 1). T$_N$ v CD95int $P=0.0005$, CD95int v CD95hi $P=0.003$, CD95hi v T$_{CM}$ $P=0.002$, T$_N$ v T$_{CM}$ $P=0.002$, paired Wilcoxon, number of independent tests=3. **$P<0.01$, ***$P<0.001$, ****$P<0.0001$. Bars represents the median, the box the interquartile range (IQR) and whiskers show lower quartile − 1.5 * IQR and upper quartile + 1.5 * IQR; all data points (including outliers) are shown by filled circles. Abbreviations: T$_N$: T naïve, T$_{SCM}$: T stem-cell memory, T$_{CM}$: T central memory, MFI: median fluorescence intensity. The data underlying this figure can be found in S1 Data.

response to all 3 conditions (CD95hi v CD95int: IL-7/IL-15 $P=0.0001$, $N$ = 13; IL-15 $P=0.0002$, $N=10$; IL-2 $P=0.1$, $N=5$. CD95hi v T$_{CM}$: IL-7/IL-15 $P=0.0008$, $N=13$; IL-15 $P=0.1$, $N=10$; IL-2 $P=0.5$, $N=5$; unpaired Wilcoxon), Fig 2A. We conclude that CD95hi T$_{SCM}$ have higher self-renewal in the context of homeostatic cytokines than CD95int T$_{SCM}$ and T$_{CM}$. The differentiation profile of the sorted cells at day 7 is given in Fig C in S1 Text and the CTV profile in Fig D in S1 Text.

In response to αCD3αCD28 stimulation, a strikingly different pattern emerged. In this scenario, both T$_{SCM}$ subpopulations rapidly differentiated whereas a large proportion of divided T$_{CM}$ cells retained their phenotype (Fig 2B). Investigating this further, we found that both T$_{SCM}$ subpopulations were able to differentiate into a number of other phenotypes: T$_{CM}$, T$_{EM}$ and T$_{EMRA}$ were all generated by both CD95hi and CD95int T$_{SCM}$ cells in appreciable quantities, Fig 2C. In contrast T$_{CM}$ were only able to produce T$_{EM}$ in measurable quantities, Fig 2C. Quantifying this as a "multipotency index" ("Materials and methods"), we found that both CD95hi T$_{SCM}$ and CD95int T$_{SCM}$ had a significantly higher multipotency index than T$_{CM}$ ($P=0.01$, $N=4$ and $P=0.0004$, $N=7$, respectively, unpaired Wilcoxon rank sum, Fig 2D).

Together, these results suggests that CD95hi T$_{SCM}$ are both better able to maintain themselves during homeostasis but also better able to generate a multicellular immune response during challenge when compared with T$_{CM}$ cells.

Next, we investigated the effector profile of the T$_{SCM}$ subpopulations following stimulation. CD95int T$_{SCM}$ and CD95hi T$_{SCM}$ populations, together with comparator T$_N$, T$_{CM}$, T$_{EM}$ and T$_{EMRA}$ cells were sorted at high purity, stimulated with αCD3αCD28 for 12 h and assayed for production of cytokines and molecules associated with cytotoxicity by flow cytometry. We found that a greater proportion of CD95hi T$_{SCM}$ cells tended to produce IFNγ, IL-2 and TNFα compared to both CD95int T$_{SCM}$ and T$_{CM}$ (Fig 2E, Fig E in S1 Text). CD95hi T$_{SCM}$ also degranulated more readily (measured by CD107a) compared with T$_{CM}$; the frequency of granzyme B and perforin-positive cells were very low in all three populations but again followed the same trend (Fig 2E, Fig E in S1 Text).

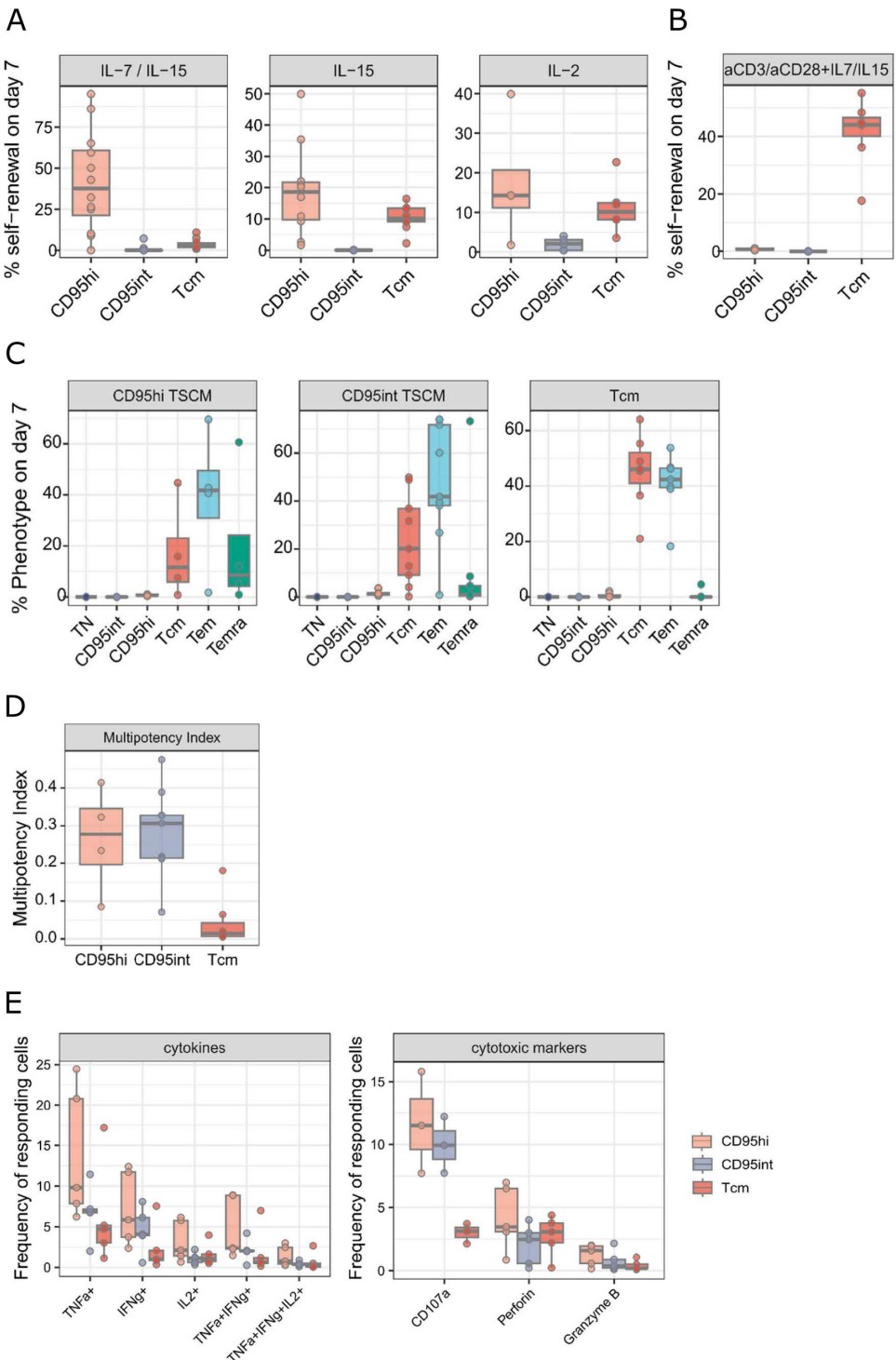

**Fig 2. CD95hi T$_{SCM}$ cells exhibit increased self-renewal and functionality compared to both CD95int T$_{SCM}$ cells and T$_{CM}$ cells. (A)** Self-renewal in response to homeostatic cytokines was investigated by culturing sorted CD95hi, CD95int and T$_{CM}$ for 7 days with 25 ng/ml IL-7/IL-15, 25 ng/mL IL-15 or 6.1 ng of IL-2 (N = 13, 10, 5, respectively). The self-renewal (defined as percentage of divided cells that have maintained their input phenotype) of each subset on day 7 is shown. The self-renewal of CD95hi T$_{SCM}$ was consistently higher than that of CD95int T$_{SCM}$ or T$_{CM}$ in response to all 3 conditions (CD95hi v CD95int: IL-7/IL-15 P = 0.0001, N = 13; IL-15 P = 0.0002, N = 10; IL-2 P = 0.1, N = 5. CD95hi v T$_{CM}$: IL-7/IL-15 P = 0.0008, N = 13; IL-15 P = 0.1, N = 10; IL-2 P = 0.5, N = 5. Unpaired Wilcoxon, number of independent tests per dataset = 2). **(B)** Self-renewal in response to TCR stimulation

was investigated by culturing sorted T cell subpopulations for 7 days with 25 ng/ml IL-7/IL-15, 2 µg/ml aCD3 and 1 µg/ml aCD28. **(C)** Multipotency in response to TCR stimulation was investigated by culturing sorted CD95hi, CD95int and $T_{CM}$ for 7 days with 25 ng/ml IL-7/IL-15, 2 µg/ml αCD3 and 1 µg/ml αCD28. The phenotype of the starting (sorted) population is shown in the grey bars above each panel and the subpopulations they differentiated into are plotted. **(D)** The data from panel C is used to calculate a multipotency index, this captures the diversity in subpopulations that a subpopulation generates and is defined for a subpopulation $j$ as $MI_j = -\sum_{\substack{i=1 \\ i \neq j}}^{n} p_i \log_{10}(p_i)$ where $p_i$ is the proportion of differentiated cells (i.e., those that have changed phenotype) that are represented by subpopulation $i$. $N = 9$ but due to low frequencies of some cell populations $N = 4$ for CD95hi $T_{SCM}$ and $N = 7$ for $T_{CM}$. **(E)** FACS-isolated T cell subsets were stimulated for 12 h with 2 µg/ml αCD3 and 1 µg/ml αCD28. Boxplots on left summarize the percentage of TNFα, IFNγ and IL-2 expressing cells among the different subsets. Boxplot on right shows markers associated with cytotoxicity. $N = 5$. An extension of panel E to show $T_N$ and $T_{EM}$ is shown in Fig E in S1 Text. In all cases bars represents the median, the box the interquartile range (IQR) and whiskers show lower quartile − 1.5 * IQR and upper quartile + 1.5 * IQR; all data points (including outliers) are shown by filled circles. All subjects from Cohort 1. A replicate of this figure but with individuals identified by different symbols is provided in Fig S in S1 Text. The data underlying this figure can be found in S1 Data.

Finally, given that two recent studies have reported heterogeneity in CD122 (IL-2Rβ) expression and PD1/TIGIT coexpression in the $T_{SCM}$ population we studied expression of these proteins in the CD95int $T_{SCM}$ and CD95hi $T_{SCM}$ subpopulations [18,19]. We found that both CD122$^+$ cells and PD1$^+$TIGIT$^+$ cells were enriched in the CD95hi $T_{SCM}$ population (Fig F in S1 Text). We further looked at the relationship between Ki67 expression and PD1/TIGIT coexpression. We found no evidence that exclusion of the "exhausted" PD1$^+$TIGIT$^+$ $T_{SCM}$ and exhausted PD1$^+$TIGIT$^+$ $T_{CM}$ cells made Ki67 expression in the remaining $T_{SCM}$ and $T_{CM}$ populations more similar (Fig G in S1 Text). So, at least when considering dynamics, our data are not consistent with the suggestion [19] that PD1$^-$TIGIT$^-$ $T_{CM}$ and PD1$^-$TIGIT$^-$ $T_{SCM}$ cells are a single homogeneous population.

**$T_{SCM}$ subsets have different gene expression profiles**

Next, we investigated gene expression by performing bulk RNAseq analysis of CD8$^+$ $T_N$, CD95int $T_{SCM}$, CD95hi $T_{SCM}$, $T_{CM}$, $T_{EM}$ and $T_{EMRA}$ populations sorted from PBMC of healthy donors. Cells were either unstimulated (cultured with RPMI) or stimulated with αCD3αCD28. RNA was extracted and sequenced ("Materials and methods"). PCA representation of the most variable genes confirmed firstly that unstimulated cells of all phenotypes tended to cluster together and away from all stimulated cells regardless of phenotype. Further, for unstimulated cells, CD95int $T_{SCM}$ clustered near $T_N$, whereas CD95hi $T_{SCM}$ clustered with $T_{CM}$. Upon stimulation these four populations separated with both $T_{SCM}$ subpopulations distinct from each other and from $T_N$ and $T_{CM}$ cells; Fig 3A. Both unstimulated $T_{EM}$ and $T_{EMRA}$ cells and stimulated $T_{EM}$ and $T_{EMRA}$ cells clustered closely with each other as has previously been noted [17,18]. For the unstimulated CD95int and CD95hi $T_{SCM}$ subpopulations we calculated which genes (after batch correction) were differentially expressed and then identified which pathways these genes were enriched in; Fig 3B. The results were very clear and self-consistent: pathways that were upregulated in naïve CD8 and CD4 T cells compared to memory T cells were upregulated in CD95int $T_{SCM}$ compared to CD95hi $T_{SCM}$; Fig 3B. The results from repeating the same analysis for the stimulated CD95int and CD95hi $T_{SCM}$ subpopulations were harder to interpret; Fig 3C. Only one of the pathways with enriched genes was a T cell pathway (GSE21360, PRIMARY versus QUATERNARY MEMORY CD8 TCELL UP from [20]). This pathway suggests that genes upregulated in CD95hi $T_{SCM}$ are enriched in genes upregulated in memory CD8 T cells following multiple antigen challenge ("QUATERNARY MEMORY CD8 TCELL") whereas genes upregulated in CD95int $T_{SCM}$ are associated with memory CD8$^+$ T cells following primary antigen challenge ("PRIMARY MEMORY CD8 TCELL"). Heatmaps showing the 70 most differentially expressed genes for both the unstimulated CD95int and CD95hi $T_{SCM}$ and the stimulated CD95int and CD95hi $T_{SCM}$ are shown in Panels (A) and (B) of Fig H in S1 Text, respectively.

We validated these conclusions by qPCR. We focused *a priori* on 9 genes known to be differentially expressed between $T_N$ and $T_{SCM}$ cells and between $T_{SCM}$ cells and $T_{CM}$ cells [1] ("Materials and methods"). We found that sets of genes associated with $T_N$ cells were more highly expressed by CD95int $T_{SCM}$ cells than CD95hi $T_{SCM}$ cells ($P = 0.012$, Binomial two tailed) while sets of genes associated with $T_{CM}$ were more highly expressed by CD95hi $T_{SCM}$ cells compared to

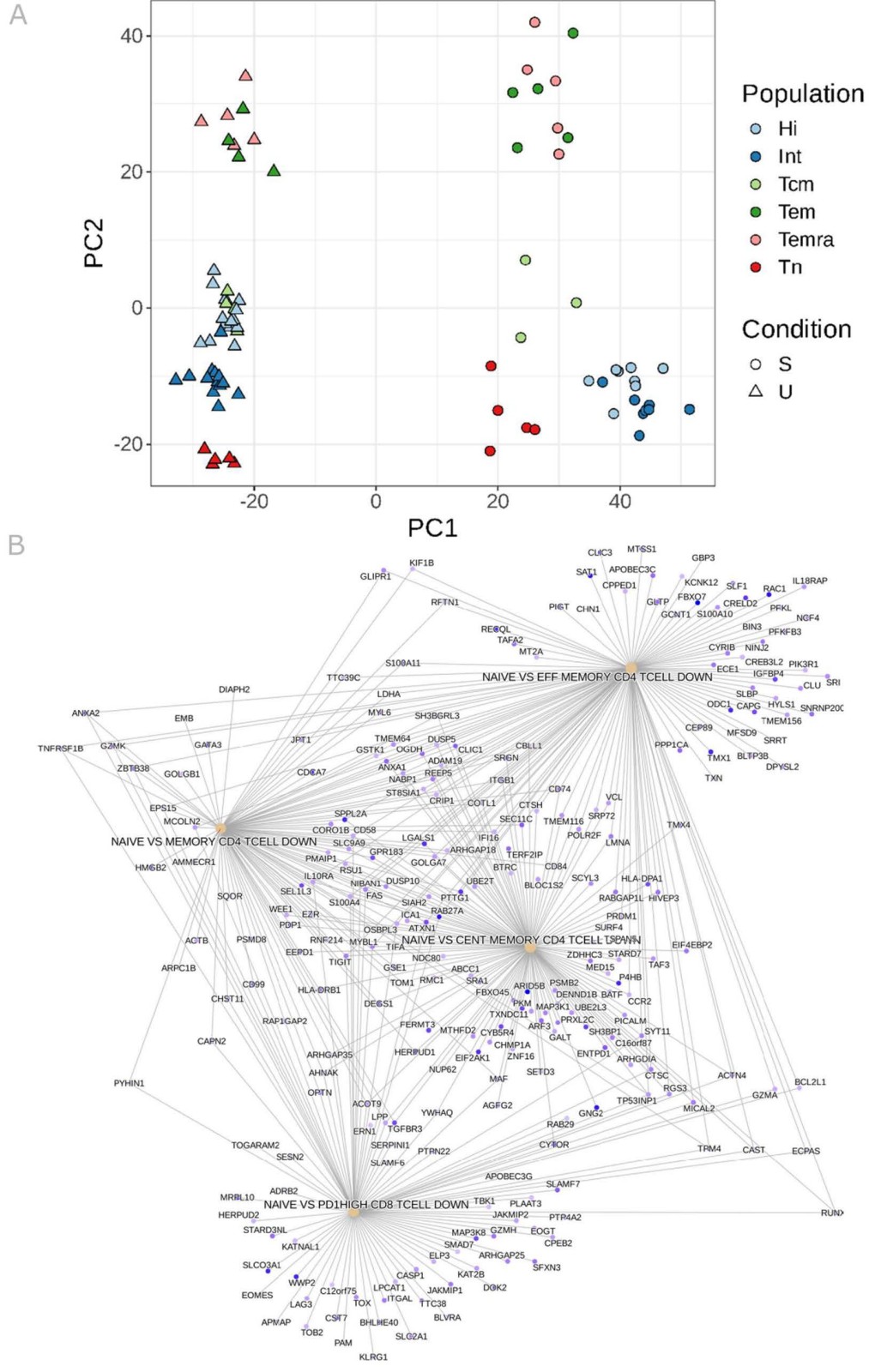

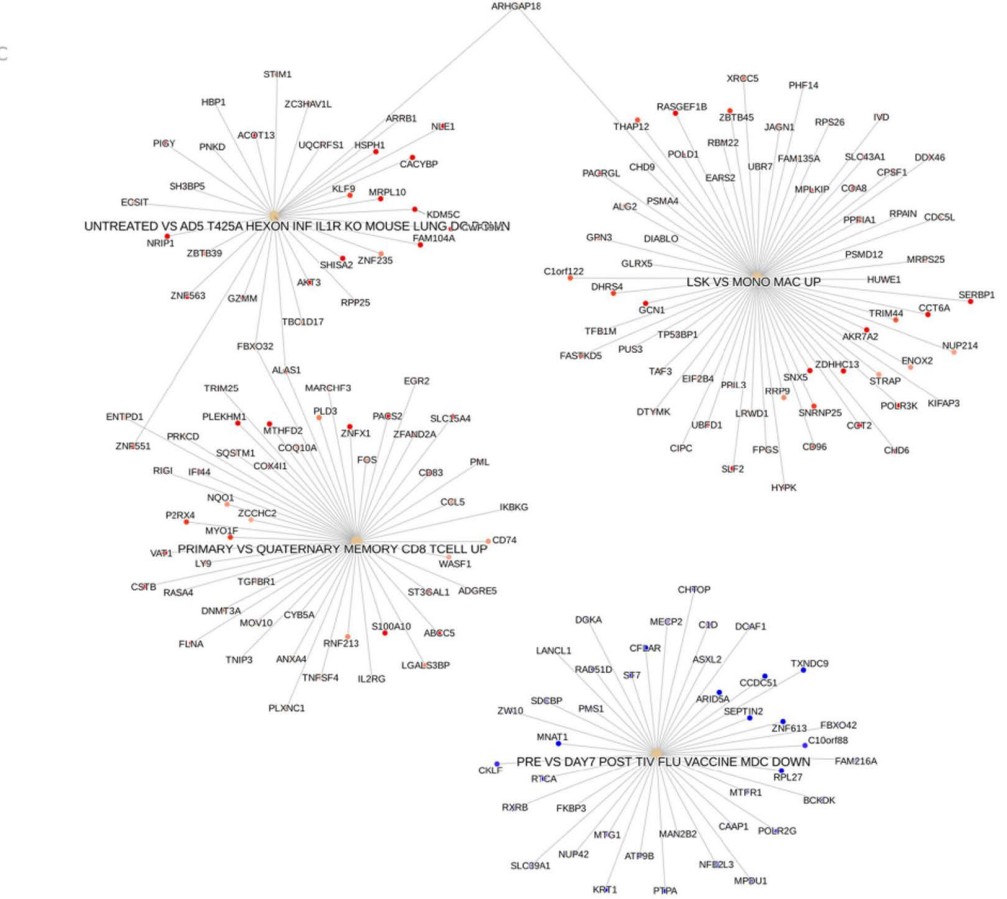

**Fig 3. Analysis of bulk RNAseq data.** CD8+ $T_N$, CD95int $T_{SCM}$, CD95hi $T_{SCM}$, $T_{CM}$, $T_{EM}$ and $T_{EMRA}$ populations were sorted. Cells were either unstimulated or stimulated with αCD3αCD28. RNA was extracted and sequenced. **(A)** PCA representation of the most variable genes showed that, for unstimulated cells, CD95int $T_{SCM}$ clustered near $T_N$, whereas CD95hi $T_{SCM}$ clustered with $T_{CM}$. Upon stimulation these four populations separated with both $T_{SCM}$ subpopulations distinct from each other and from $T_N$ and $T_{CM}$ cells. **(B)** Gene set enrichment analysis of genes differentially expressed between unstimulated CD95int $T_{SCM}$ and unstimulated CD95hi $T_{SCM}$ found, transcriptionally CD95int $T_{SCM}$ were closer to naïve CD4 and CD8+ T cells while CD95hi $T_{SCM}$ were correspondingly closer to various memory populations (i.e., genes that were down in naïve v memory CD4, naïve v effector memory CD4 and naïve v PD1 hi CD8 were also down in CD95int $T_{SCM}$). **(C)** The same analysis for of genes differentially expressed between stimulated CD95int $T_{SCM}$ and stimulated CD95hi $T_{SCM}$ was harder to interpret. The only T cell pathway (GSE21360, PRIMARY VS QUATERNY MEMORY CD8 TCELL UP) implies that genes upregulated in CD95hi $T_{SCM}$ are enriched in genes upregulated in memory CD8 T cells following multiple antigen challenge whereas genes upregulated in CD95int $T_{SCM}$ are associated with memory CD8+ T cells following primary antigen challenge.

CD95int $T_{SCM}$ cells ($P = 0.015$, Binomial two tailed), Panels (A) and (B) of Fig I in S1 Text, respectively. Given these trends in the data, we investigated canonical differentiation markers to see if there was a trend from $T_N$ to CD95int, to CD95hi, to $T_{CM}$. We quantified CCR7 and CD45RA (protein) expression by flow cytometry and, consistent with the gene profiles, found that for both CCR7 and CD45RA, expression was highest in $T_N$, significant lower in CD95int than $T_N$ ($P = 1 \times 10^{-6}$, $P = 1 \times 10^{-6}$), significantly lower in CD95hi than CD95int ($P = 1 \times 10^{-6}$, $P = 0.0006$) and significantly lower in $T_{CM}$ than in CD95hi ($P = 0.0006$, $P = 1 \times 10^{-6}$); i.e., there was a strong and consistent monotonic decrease in CCR7 and CD45RA expression from $T_N$ through CD95int, CD95hi to $T_{CM}$ (Panels (C) and (D) of Fig I in S1 Text)); $N = 21$ two-tailed paired Wilcoxon, number of independent tests ≤6.

PLOS Biology

### CD95int and CD95hi $T_{SCM}$ subsets have different lymph node homing properties

Different T cell subpopulations have different tissue homing properties related to their function [21]. We investigated whether the distribution of CD95int and CD95hi $T_{SCM}$ cells varied between blood and lymph node after immune challenge by intramuscular injection with a model antigen. Lymph node cells were collected by ultrasound guided fine needle aspiration of axillary lymph nodes from 7 healthy volunteers participating in an experimental synthetic HIV envelope immunogen study (reported elsewhere [22]); matched blood samples from the same individuals at the same time point were also collected. Cells were collected between 7 days and 56 days post-first immunization but before the subsequent boost, a time point at which antigen-specific T cells but not antigen-specific IgG was detectable [22]. We found that the percentage of $T_{SCM}$ cells that were CD95int was significantly higher in the lymph node than in the blood (median 81.75% in LN versus 71.8% in PBMC, $P=0.029$, unpaired Wilcoxon) and conversely that the percentage of $T_{SCM}$ that were CD95hi tended to be elevated in blood (median 11.6% in LN versus 23.7% in PBMC, $P=0.079$, unpaired Wilcoxon), Fig 4.

### $T_{SCM}$ subsets have different division histories and different long-term stability

T cell receptor excision circles (TRECs) are small circles of DNA created by DNA excision during V-(D)-J gene rearrangement in the thymus. TRECs are only produced upon T cell receptor rearrangement and are not replicated when a cell divides so they are diluted upon mitosis. TREC content is therefore an indicator of the number of division events that a T

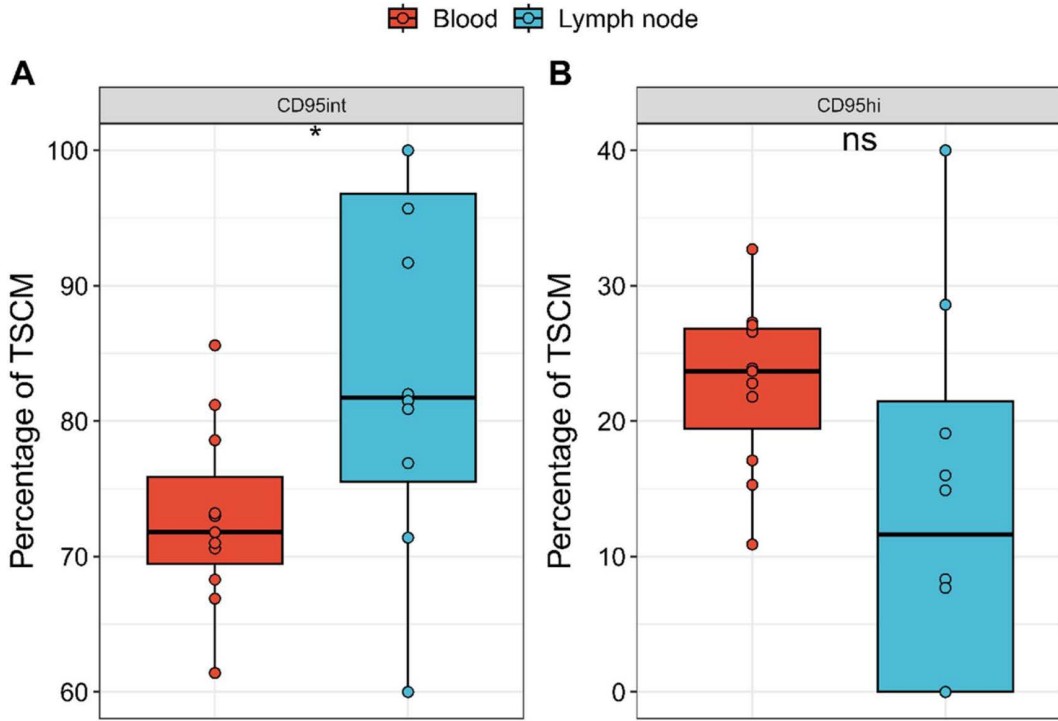

**Fig 4. CD95hi and CD95int frequencies differ between the blood and the lymph nodes.** PBMCs and lymph node cells isolated from the blood and axillary lymph nodes, respectively, of 7 healthy vaccinated volunteers from Cohort 3 were isolated and stained. Cells were gated on the population of **(A)** CD95int cells and **(B)** CD95hi cells. PBMC shown in red and lymph node cells in blue. Each dot represents one sample. CD95int $T_{SCM}$ (as a proportion of total $T_{SCM}$) are significantly enriched in the LN, $P=0.029$ unpaired two-tailed Wilcoxon test. Conversely CD95hi $T_{SCM}$ cells show a non-significant trend to be enriched in the blood $P=0.079$. *$P < 0.05$. Bars represents the median, the box the interquartile range (IQR) and whiskers show lower quartile − 1.5 * IQR and upper quartile + 1.5 * IQR; all data points (including outliers) are shown by circles. The data underlying this figure can be found in S1 Data.

cell population has undergone since thymic production. TREC content is typically very low in memory T cells and difficult to quantify by qPCR or even digital droplet PCR (ddPCR) from extracted DNA. We therefore developed a highly sensitive ddPCR assay directly from cell lysate [23]. Using this assay, we quantified the TREC content in high purity FACS sorted populations of CD8+ $T_N$, CD95int $T_{SCM}$, CD95hi $T_{SCM}$, $T_{CM}$, $T_{EM}$ and $T_{EMRA}$ (Fig 5A). As expected, TREC content was higher in $T_N$ cells compared to memory cells. Furthermore, the TREC content of CD95int $T_{SCM}$ cells was significantly higher than that of CD95hi $T_{SCM}$ cells indicating that these are distinct populations with different division histories (if there was frequent interconversion between CD95hi $T_{SCM}$ and CD95int $T_{SCM}$ then these populations would have similar TREC contents) and furthermore that CD95hi $T_{SCM}$ cells have undergone more rounds of division since thymic production than CD95int $T_{SCM}$ cells.

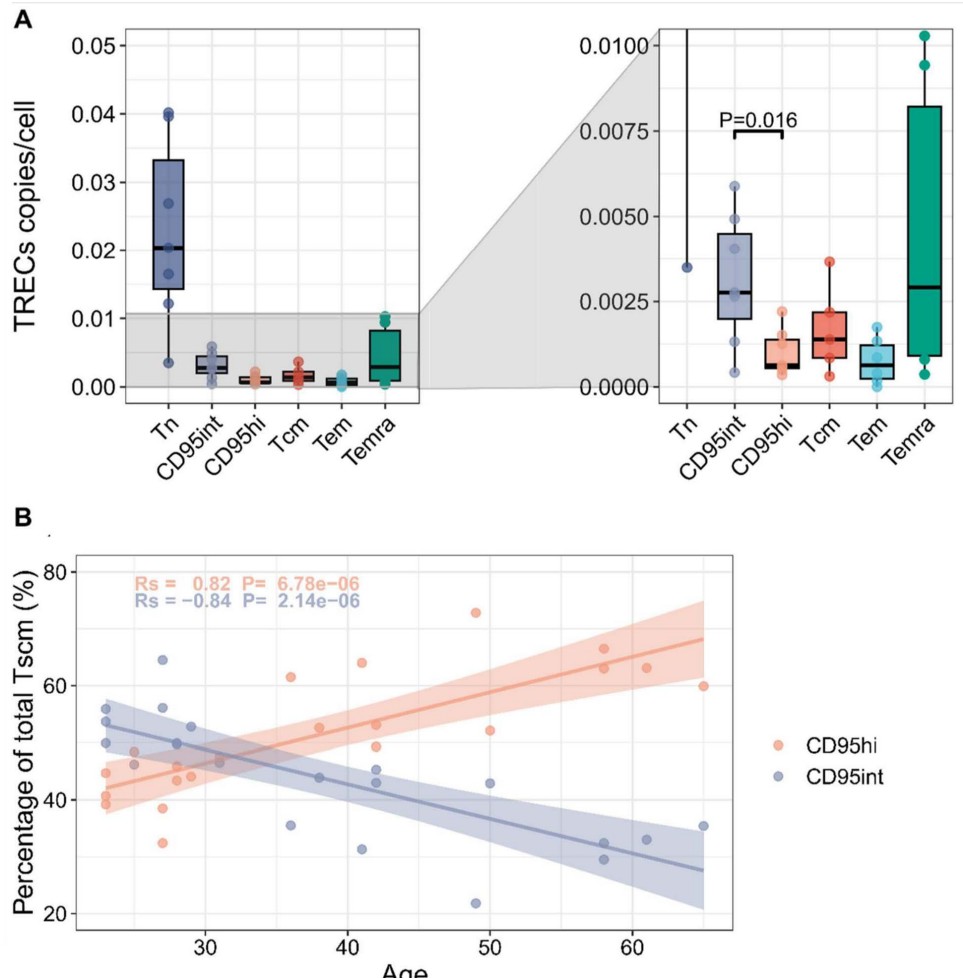

**Fig 5. CD95hi cells have a lower TREC content and their frequency increases with age. (A)** TREC (T-cell receptor excision circle) content was quantified by ddPCR for FACS-sorted T cell subpopulations from $N = 7$ healthy donors, from Cohort 4. Boxplots indicate the number of TREC copies per cell for each subset (bars represents the median, the box the interquartile range (IQR) and whiskers show lower quartile − 1.5 * IQR and upper quartile + 1.5 * IQR; all data points, including outliers, are shown by circles). Graph on right is a zoom in (truncated $y$ axis) to show TREC content of memory populations more clearly. Statistics: Wilcoxon paired test **(B)** PBMCs isolated from the blood of 21 healthy individuals (23−65 yrs, Cohort 4) were stained to identify CD95hi and CD95int $T_{SCM}$ cells. The percentage of $T_{SCM}$ that were CD95hi (orange circles) and CD95int (blue circles) was quantified. The results are plotted against the age of the donor. We found that the proportion of $T_{SCM}$ that were CD95hi increased significantly with age (Rs = 0.82, $P = 6.78 \times 10^{-6}$ Spearman). The data underlying this figure can be found in S1 Data.

Next, we examined the change in CD95int $T_{SCM}$ and CD95hi $T_{SCM}$ frequency with donor age. We quantified the frequency of the two subpopulations in a cross-sectional cohort of 21 individuals whose ages ranged from 23 to 65 yrs. We found that the frequency of the $T_{SCM}$ population that was CD95hi increased significantly with age (Rs = 0.82, $P = 7 \times 10^{-6}$, Spearman correlation) while the CD95int $T_{SCM}$ population showed a corresponding decrease (Rs = −0.84, $P = 2 \times 10^{-6}$, Spearman correlation), Fig 5B.

**CD95int and CD95hi $T_{SCM}$ subsets have different dynamics in humans *in vivo***

Finally, we investigated the *in vivo* life span and lineage relationship of CD95int $T_{SCM}$ cells and CD95hi $T_{SCM}$ cells using stable isotope labelling. Three individuals received 7 weeks of deuterated "heavy" water ($2H_2O$, "Materials and methods"); serial saliva and blood samples were taken during and up to 140 days after the start of labelling, Fig 6A. Monocytes were extracted as a rapidly turning over reference population ("Materials and methods", Figs J–K in S1 Text, Table D in S2 Text). The remaining PBMCs were FACS sorted into CD8$^+$ $T_N$, CD8$^+$ CD95int $T_{SCM}$ and CD8$^+$ CD95hi $T_{SCM}$ subpopulations. Deuterium enrichment in the saliva as well as in the DNA of monocytes and sorted T cell subpopulations was measured by gas chromatography/mass spectrometry (GC/MS) [24], Fig 6B. Strikingly, the stable isotope enrichment for the two $T_{SCM}$ subpopulations remained distinct for the duration of the labelling experiment (140 days), indicating that these are distinct subpopulations on this timescale.

We constructed 5 different mathematical models encapsulating five different assumptions about the lineage relationship between the populations (Fig 7, "Materials and methods") and sought to identify which model and which parameter combinations were most likely to have generated the observed data. In order to further constrain the models and to improve parameter identifiability we fitted two further data sets we had previously obtained [4,12]: one measured the telomere length in $T_N$ and total $T_{SCM}$ cells from 5 individuals and one quantified the frequency of yellow fever virus (YFV)-specific total $T_{SCM}$ in a cross sectional cohort of 37 individuals at different time points (ranging from 0.27 to 35.02 yrs) post-YF vaccination. The three datasets were fitted simultaneously. Plots of best fits are given in Fig 8 and Figs L–O in S1 Text with the corresponding parameters in Table 1 in main text and Tables F–I in S2 Text. We found that three of the five models gave good fits: the model with a forked structure in which a fraction of $T_N$ differentiated into CD95hi $T_{SCM}$ and a fraction into CD95int $T_{SCM}$ (Fig 7C) and the two linear models (Fig 7D and 7E) (Table K in S2 Text). The independent homogenous and heterogeneous models (Fig 7A and 7B) were not viable as they failed to capture the decline in the YFV-specific population (Panel (C) of Fig L and Panel (C) of Fig M in S1 Text). We conclude that, of the models we considered, the fork model and the two linear models were consistent with the data and could be considered viable options at this stage; the two independent models could be ruled out.

We calculated the clonal life span of CD95int and CD95hi $T_{SCM}$ cells implied by the three viable models (Table 1 in main text, Tables H and I in S2 Text). All three models indicated that the CD95int $T_{SCM}$ population was very long-lived at the clonal level (half-life of a CD95int $T_{SCM}$ clone of the order of 10 yrs) whereas the CD95hi population was shorter-lived (half-life of a CD95hi $T_{SCM}$ clone of the order of months), Fig 9. We conclude that the CD95int $T_{SCM}$ subset is most likely to be the population responsible for retaining long-term immune memory.

**Model predictions agree well with an unseen data set**

To validate the models and parameters found in the previous section we assessed their ability to predict the long-term dynamics of YFV-specific CD95int $T_{SCM}$ and CD95hi $T_{SCM}$ obtained from Fuertes Marraco and colleagues [2]. The flow panels used to analyze the YFV-specific $T_{SCM}$ dataset in Fuertes Marraco and colleagues did not systematically include CD95 so we were not able to explicitly quantify the frequency of YFV-specific CD95int and CD95hi $T_{SCM}$ cells over time. However, given the strong association between CD45RA, CCR7 and CD95 we were able to use CD45RA and CCR7 expression to distinguish the two $T_{SCM}$ subpopulations (Panel (C) and (D) of Fig I in S1 Text, Fig P in S1 Text). We thus use

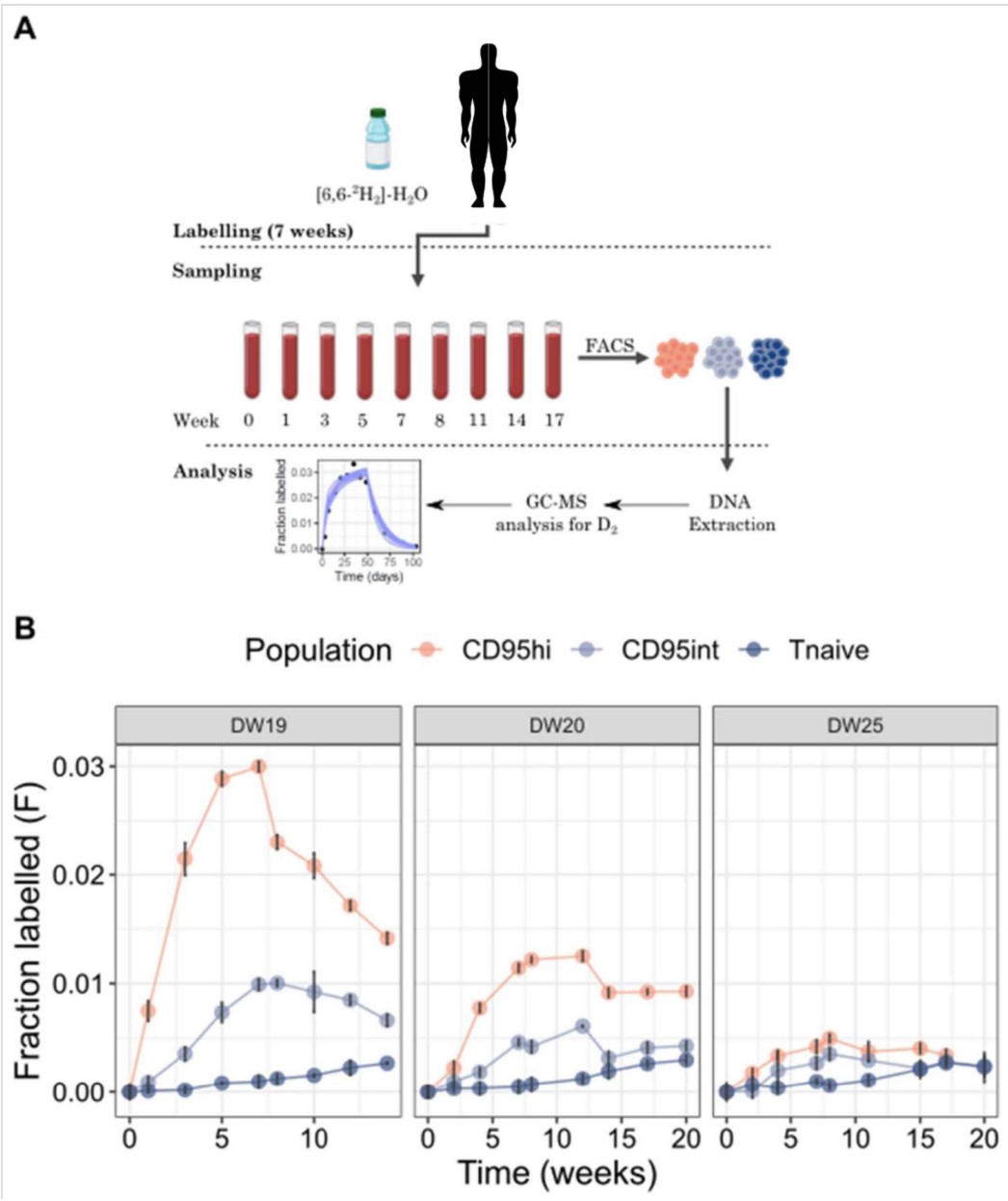

**Fig 6. The CD95hi subset incorporated more label *in vivo* compared to CD95int. (A)** Workflow of heavy water labelling technique and timepoints of sample collection. Participants from Cohort 5 drank 70% deuterated water for 7 weeks (100 ml twice daily for one day, then 50 ml twice daily thereafter). Saliva and PBMC were collected at multiple time-points during and after labelling. Cells of interested were sorted and deuterium enrichment was measured by gas chromatography/mass spectrometry **(B)** M + 1 enrichment of DNA from selected cell populations is shown per individual for all time-points for the three different subsets. Error bars represent analytic variance and show the standard deviation of 4 replicate measurements. Note, DW19 attained higher levels of label in body water hence the higher enrichment in DNA (the mathematical modelling allows for this variation when calculating cell kinetics). Here lines connect neighboring timepoints for ease of viewing, they do not represent model fits. The data underlying this figure can be found in S1 Data.

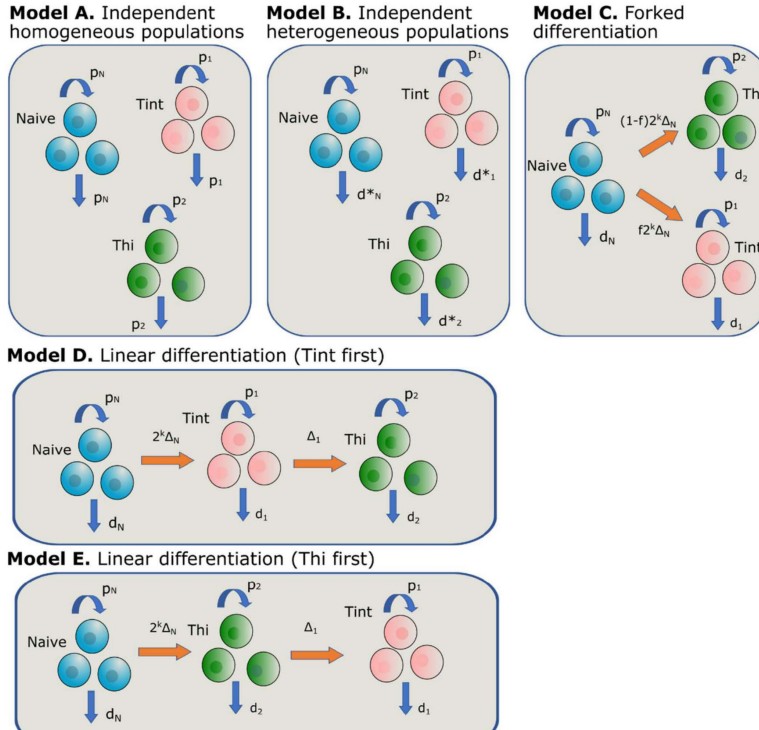

**Fig 7. Models capturing different kinetics and relationships between the cell populations of interest. (A) Independent homogeneous model.** Each population ($T_N$, CD95int, CD95hi) is independent and homogeneous. **(B) Independent heterogeneous model.** Each population ($T_N$, CD95int, CD95hi) is independent and allowed (but not constrained) to be heterogeneous. **(C) Forked differentiation model.** Upon meeting cognate antigen, $T_N$ cells undergo a clonal burst (of size $2^k$); a fraction (f) differentiate into the CD95int pool with the remainder differentiating into the CD95hi pool. **(D) Linear differentiation model (CD95int first).** Upon meeting cognate antigen, $T_N$ cells undergo a clonal burst (of size $2^k$) and differentiate into the CD95int pool which subsequently differentiate into the CD95hi pool. **(E) Linear differentiation model (CD95hi first).** Upon meeting cognate antigen, $T_N$ cells undergo a clonal burst (of size $2^k$) and differentiate into the CD95hi pool which subsequently differentiate into the CD95int pool. Note for compactness, in the cartoons, CD95int $T_{SCM}$ are designated Tint and CD95hi $T_{SCM}$ are designated Thi. In every model each population was assumed to be of constant size. The models were fitted to the data to identify which model was most consistent with the experimental observations and to quantify the population parameters.

CCR7++CD45RA++ cells ("naïve-hi" cells in the nomenclature of [2]) as a surrogate for CD95int $T_{SCM}$ cells and the remaining CD45RA+CCR7+ cells ("naïve-int" cells in the nomenclature of [2]) as a surrogate for CD95hi $T_{SCM}$ cells. This new data set was not fitted. We simply predicted the time course of YFV-specific CD95int and CD95hi $T_{SCM}$ using the three viable models (the fork model, the linear model with $T_N$ differentiating first into CD95int $T_{SCM}$ and then into CD95hi $T_{SCM}$ and the linear model with $T_N$ differentiating first into CD95hi $T_{SCM}$ and then into CD95int $T_{SCM}$) and the best fit parameters previously identified and compared them to the new data; it should be noted that the previously fitted data included the time course of the total YFV-specific $T_{SCM}$ population and so the fitted and the new datasets are not completely independent. The agreement between the prediction and the new data was extremely good for the linear model with CD95int first (Fig 10) but unrealistic for the fork model and the linear model with CD95hi first as they both predicted a very rapid decline in the frequency of CD95hi $T_{SCM}$ that was not compatible with the observations, (Figs Q–R in S1 Text, Table L in S2 Text). We conclude that the data are best described by the linear (Tint first) model (Model D in Fig 7).

Studying the same data in a model-independent way, following the approach of Fuertes Marraco and colleagues [2], we see that the YFV-specific CCR7++CD45RA++ population (proxy for YFV-specific CD95hi $T_{SCM}$ cells) decline significantly more rapidly than the proxy for YFV-specific CD95int $T_{SCM}$ cells, Fig 11; a striking validation of our predictions.

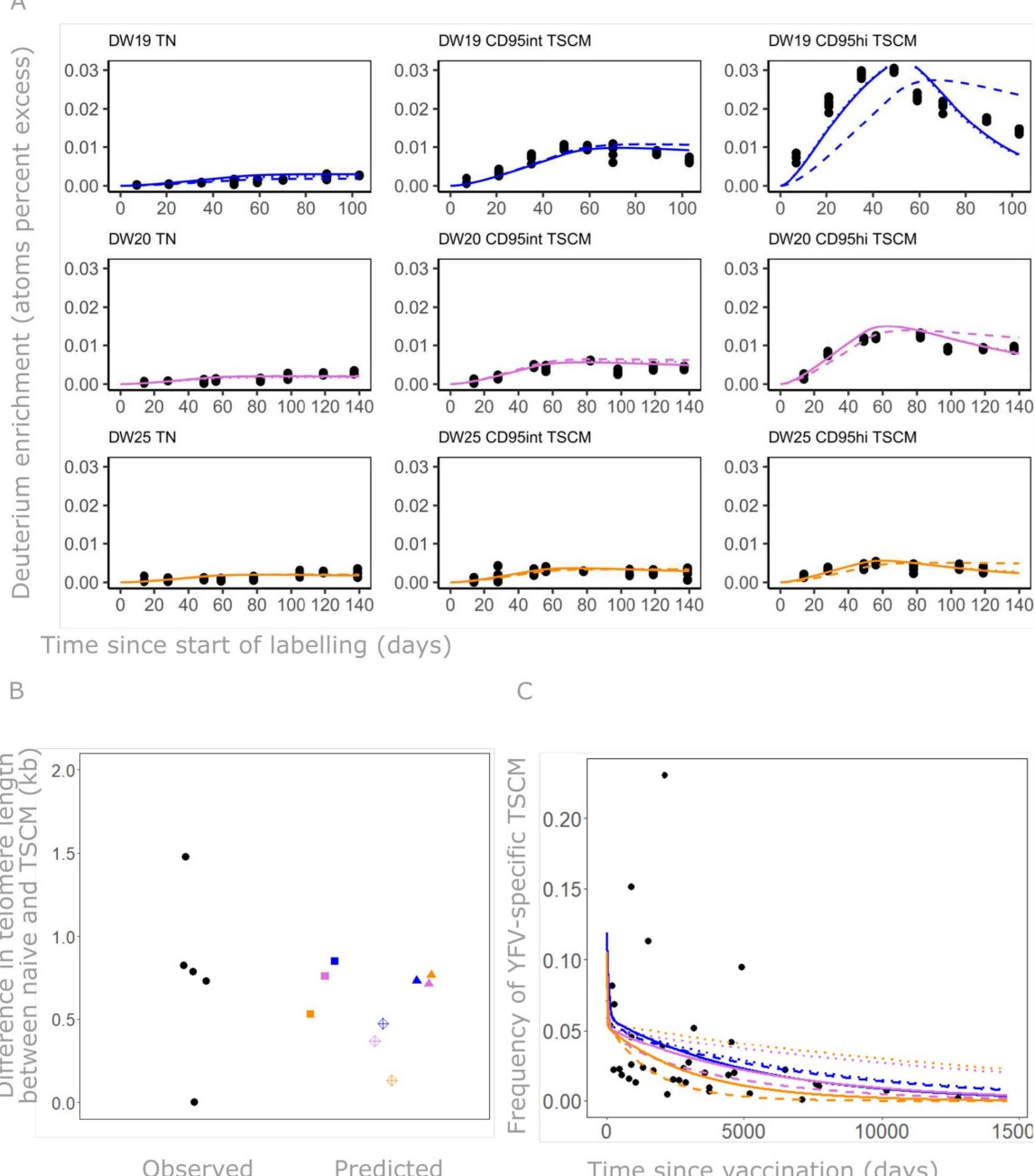

**Fig 8. Fit of the three viable models – Model C (fork), Model D (linear, CD95int first) and Model E (linear, CD95hi first) – to all data (labelling, telomere, YFV) simultaneously. (A)** Fit of models to the labelling data (each individual in a different row, each cell population in a different column); black dots represent data, colored lines represent fitted model predictions (solid: Model C, dashed: Model D, dotted: Model E). **(B)** Fit of models to the telomere data. Observed differences in mean telomere length between $T_N$ and $T_{SCM}$ for 5 individuals shown on the left in black, predictions for each individual and each model shown by colored points on the right (DW19 in blue, DW20 in pink, DW25 in orange. squares: Model C, diamonds: Model E, triangles: model D). **(C)** Fit of the model to the YFV data. Observed data (YFV-specific $T_{SCM}$ as a % of CD8+ T cells) shown by black points, fitted model predictions for each individual shown by colored lines (DW19 in blue, DW20 in pink, DW25 in orange) and for each model shown by line type (solid: Model C, dashed: Model D, dotted: Model E). The data underlying this figure can be found in S1 Data.

**Table 1. Best fit estimates of parameters obtained by fitting the winning model (Model D, linear model with CD95int T$_{SCM}$ first) to the labelling, telomere and YFV data.** Interquartile range (estimated by bootstrapping the data) shown in square brackets below the corresponding estimates. Note, the clonal half-life corresponds to the half-life of the clone (ln2/d-p) not the half-life of the cell as is typically reported (ln2/d). Best fit estimates for the other 4 models are provided in Tables G–J in S2 Text. We found that CD95int TSCM cells proliferated much more slowly than CD95hi TSCM cells with an average proliferation rate of 0.0008 per day (equivalent to one division every 3.4 yrs) whereas CD95hi TSCM cells had an average proliferation rate of 0.002 per day (equivalent to one division every 1.4 yrs). The data in this table can be found in S1 Data.

| Id | $p_N$ (day⁻¹) | $p_1$ (day⁻¹) | $p_2$ (day⁻¹) | $\Delta N$ (day⁻¹) | $\Delta_1$ (day⁻¹) | $k$ | $A$ (% of CD8) | Clonal half-life CD95int (days) | Clonal half-life CD95hi (days) |
|---|---|---|---|---|---|---|---|---|---|
| DW19 | 0.00023 [0.00022, 0.00034] | 0.0013 [0.0011, 0.0019] | 0.004 [0.004, 0.06] | $1 \times 10^{-6}$ [$2 \times 10^{-9}$, $1 \times 10^{-6}$] | 0.001 [0.001, 0.008] | 1.8 [1.3, 10.0] | 0.04 [0.03, 0.04] | 5231 [3717, 18206] | 194 [37, 209] |
| DW20 | 0.00031 [0.00028, 0.00038] | 0.0008 [0.0006, 0.0008] | 0.002 [0.002, 0.046] | $5 \times 10^{-9}$ [$2 \times 10^{-10}$, $2 \times 10^{-7}$] | 0.001 [0.001, 0.005] | 10.3 [4.0, 15.0] | 0.03 [0.03, 0.04] | 5409 [3213, 12307] | 590 [107, 633] |
| DW25 | 0.00032 [0.00030, 0.00034] | 0.0003 [0.0002, 0.0004] | 0.001 [0.001, 0.001] | $4 \times 10^{-10}$ [$4 \times 10^{-10}$, $8 \times 10^{-10}$] | 0.001 [0, 0.001] | 15.0 [15.0, 15.0] | 0.03 [0.03, 0.03] | 2812 [1856, 3567] | 1346 [1225, 3338] |
| **MEDIAN** | **0.00031** | **0.0008** | **0.002** | **$5 \times 10^{-9}$** | **0.001** | **10.3** | **0.03** | **5231** | **590** |

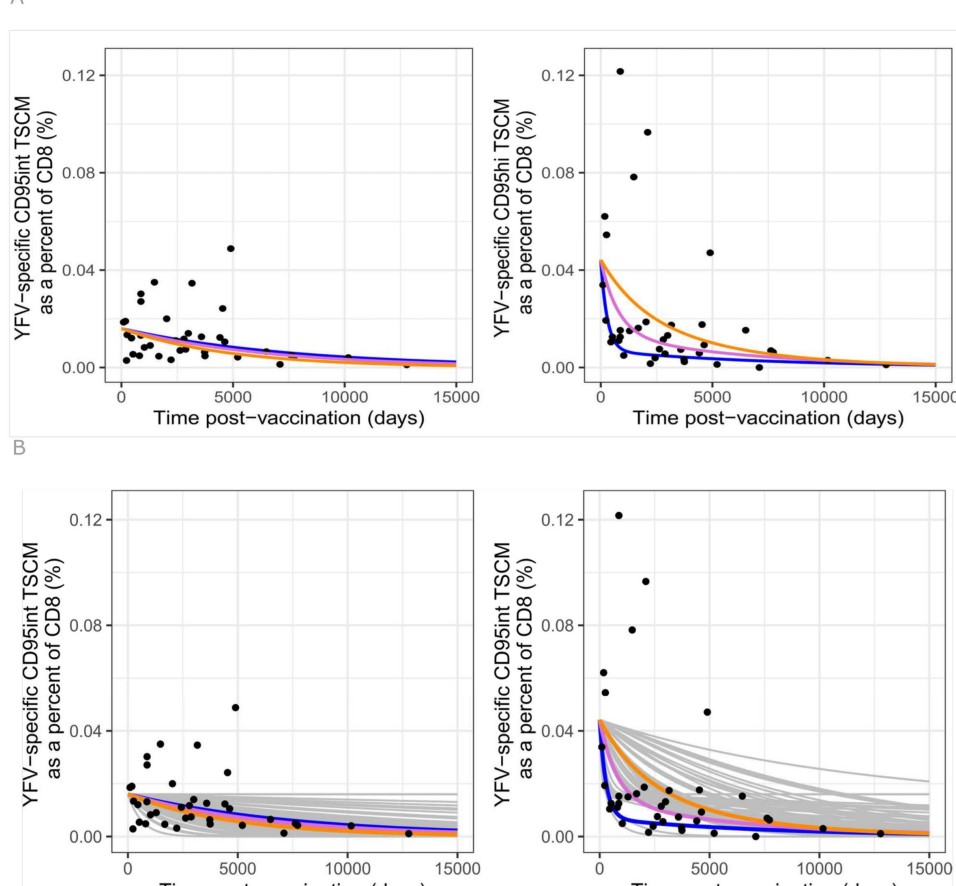

**Fig 9. Predictions of winning model – Model D (linear, CD95int first) overlaid (NOT fitted) onto a novel data set. (A)** The winning model and best fit parameters were used to predict the long-term dynamics of YFV-specific CD95int T$_{SCM}$ (left) and CD95hi T$_{SCM}$ (right) for each individual (shown by colored lines: DW19 in blue, DW20 in pink, DW25 in orange). For CD95int the predictions overlay and only one line can be seen. The predictions are plotted against but not fitted to a proxy for CD95int and CD95hi T$_{SCM}$ frequencies measured in the cross-sectional cohort post-vaccination (the same cohort for which total T$_{SCM}$ were fitted). **(B)** As above but also including 100 predictions obtained by fitting bootstrapped data (sampled with replacement) shown as grey lines. A comparison of all models to this unseen data set is provided in Table L in S2 Text. The data underlying this figure can be found in S1 Data.

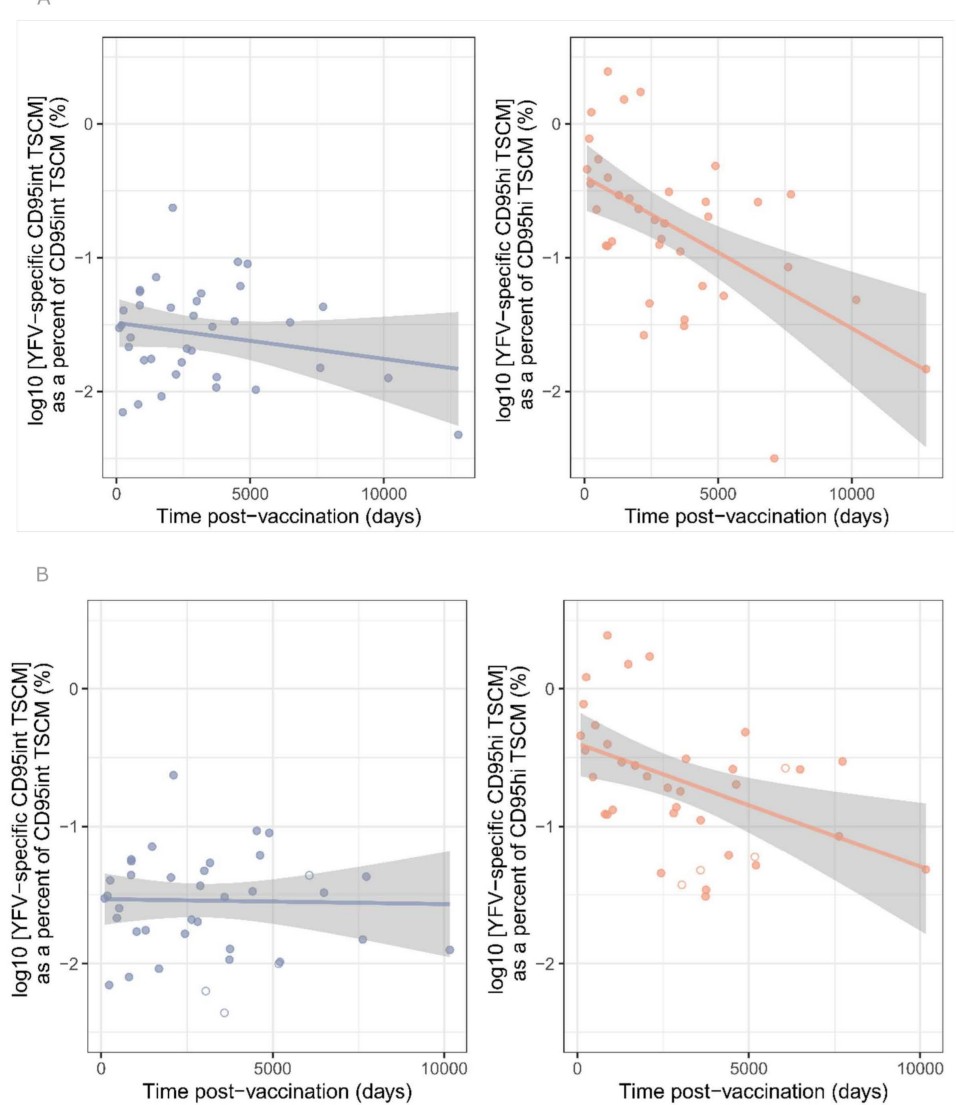

**Fig 10. A population of YFV-specific cells enriched for CD95int T$_{SCM}$ declines much more slowly over time than a population enriched for CD95hi T$_{SCM}$.** Here we reproduce the analysis from Fuertes Marraco and colleagues [2] who investigated the decline of YFV-specific memory subpopulations with time post-vaccination in a cross-sectional cohort. Fuertes Marraco and colleagues note that there is large between-individual variation in the sizes of CD8$^+$ memory subpopulations and that this noisy interindividual variation can be reduced by quantifying the frequency of YFV-specific cells within each subpopulation rather than within the total CD8$^+$ population (e.g., quantify YFV-specific T$_{CM}$ CD8$^+$ cells as a percent of total T$_{CM}$ rather than as a percent of total CD8$^+$ T cells) **(A)**. Analysis of the raw data shows that a proxy for YFV-specific CD95int T$_{SCM}$ cells (YFV-specific "naïve hi" cells) – left panel – decline significantly more slowly than a proxy for YFV-specific CD95hi T$_{SCM}$ cells (YFV-specific "naïve int" cells) – right panel ($P = 0.039$, linear regression). The blue and orange lines are the best fit straight lines to the CD95int and CD95hi T$_{SCM}$ data, respectively. **(B)**. Analysis in Fuertes Marraco and colleagues [2] differs slightly in that they chose to exclude three individuals with very low frequencies of YFV-specific CD8$^+$ T cells and to include 4 individuals who had received multiple YF vaccinations. Our conclusions are robust to these two steps (if anything they appear to strengthen the conclusion that YFV-specific CD95int T$_{SCM}$ cells decline more slowly than YFV-specific CD95hi T$_{SCM}$ cells). Recipients of multiple vaccinations shown by open symbols.

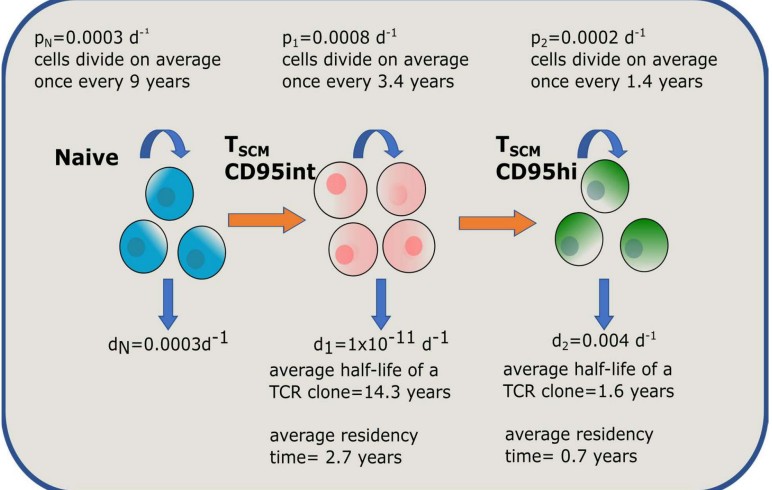

**Fig 11. Schematic of the best performing model and parameters.** An overview of the model and parameters that provides the best description of the data sets. Average residency time is average time a cell spends in that population ("Materials and methods").

This work on a new data set, which was unseen during the fitting process, corroborates the parameterized linear model and provides compelling evidence that YFV-specific CD95hi $T_{SCM}$ cells have significantly more rapid disappearance kinetics than CD95int $T_{SCM}$ cells.

## Discussion

This study establishes that the human T stem cell population consists of two distinct cell subpopulations which can be distinguished by the level of their CD95 expression. We show that these two populations have different functional profiles *ex vivo*, different transcriptional patterns, and different tissue distributions consistent with their differential expression of the lymph node homing receptor CCR7 [25]. Most importantly for the maintenance of memory, we show that there are considerable differences in the dynamics of the subpopulations whether measured by TRECs, Ki67 or stable isotope labelling with CD95hi $T_{SCM}$ proliferating significantly more rapidly than CD95int $T_{SCM}$ cells. Quantifying the labelling data, we found that CD95int $T_{SCM}$ tend to harbor very long-lived clones compared to CD95hi $T_{SCM}$; this was confirmed by the rapid decline of YFV-specific CD95hi $T_{SCM}$ post-vaccination compared with CD95int $T_{SCM}$. Crucially, using *in vivo* labelling of human volunteers, we establish that these are long-term distinct populations rather than transient states of the same population.

As well as being distinct from one another, CD95int $T_{SCM}$ cells and CD95hi $T_{SCM}$ cells are distinct from all other T cell populations. CD95hi $T_{SCM}$ cells are most similar to $T_{CM}$ but there are numerous important differences. Pivotal to the function of a stem cell population is its ability to proliferate without the whole population undergoing differentiation and phenotype change – in which case the parent population is "lost". This property is known as self-renewal. Under homeostatic conditions *ex vivo*, CD95hi $T_{SCM}$ cells have significantly higher levels of self-renewal than $T_{CM.}$ Conversely, under conditions of TCR activation, these relationships swap with CD95hi $T_{SCM}$ cells freely differentiating while $T_{CM}$ retain their phenotype (Fig 2). CD95hi $T_{SCM}$ cells therefore have significantly higher multipotency index in response to TCR stimulation than $T_{CM}$. This suggests a division of labor with CD95hi $T_{SCM}$ cells responsible for maintaining the memory population during homeostasis but during rechallenge it is CD95hi $T_{SCM}$ cells rather than $T_{CM}$ cells that rapidly differentiate and provide the required effector populations. CD95int cells are most similar to $T_N$ but again, there are considerable differences in Ki67 expression, cytokine secretion, TREC content and *in vivo* division (Figs 1, 2, 5–6). Their slow proliferation rates, very long

survival times and lower expression of CD95 may protect them from attrition thus preserving these previously antigen-selected clones for the long-term.

CD95int $T_{SCM}$ have a transcriptional profile more similar to $T_N$ while CD95hi $T_{SCM}$ more closely resemble $T_{CM}$. However, the simple assumption that CD95int and CD95hi $T_{SCM}$ cells are on a functional continuum from $T_N$ cells to more highly differentiated $T_{CM}$ cells is undermined by a number of features that do not vary monotonically across the 4 populations (Fig 12). $T_{CM}$ and $T_N$ have lower effector profiles than either $T_{SCM}$ subpopulation (Fig 2E). The ability to maintain phenotype upon proliferation does not vary monotonically either, being higher for CD95hi $T_{SCM}$ than either CD95int $T_{SCM}$ or $T_{CM}$ (Fig 2A). Similarly, Ki67 expression is not monotonic: it is low for $T_N$ and CD95int $T_{SCM}$ cells, increases dramatically for CD95hi cells and then drops again considerably for $T_{CM}$ cells (Fig 1B).

RNAseq is an extremely powerful tool but it can be problematic for identifying novel subpopulations as it is difficult to tell whether two clusters are different states of the same cell population or two distinct populations. For example, it is well known that cells from disparate cell populations will cluster together and away from their parent population when in cycle. Conversely, the same populations under different conditions can be widely separated (e.g., our Fig 3A). Therefore, existence of a separate cluster does not necessarily mean a separate subpopulation (i.e., there can be false positives) and conversely, failure to cluster does not necessarily mean two cells are from a different subpopulation (i.e., there can be false negatives). Similarly, clusters within other high dimensional datasets, such as those generated by CyTOF, can also be hard to interpret. An unbiased study of the CD8$^+$ T cell compartment revealed that, on the basis of a principal components analysis of CyTOF data, there were over 200 CD8$^+$ T cell subpopulations [26]. Likewise, a CyTOF-based study of the NK cell compartment found 6,000–30,000 different populations [27]. Given the sheer number of subpopulations identified, it seems likely that at least some of these represent multiple states of the same populations rather than hundreds (or in the case of NK cells, thousands) of long-term distinct populations. Here, by using *in vivo* labelling, we establish that CD95int $T_{SCM}$ and CD95hi $T_{SCM}$ are not transient states of the same cell population but instead are two distinct populations that remain distinct for at least 140 days.

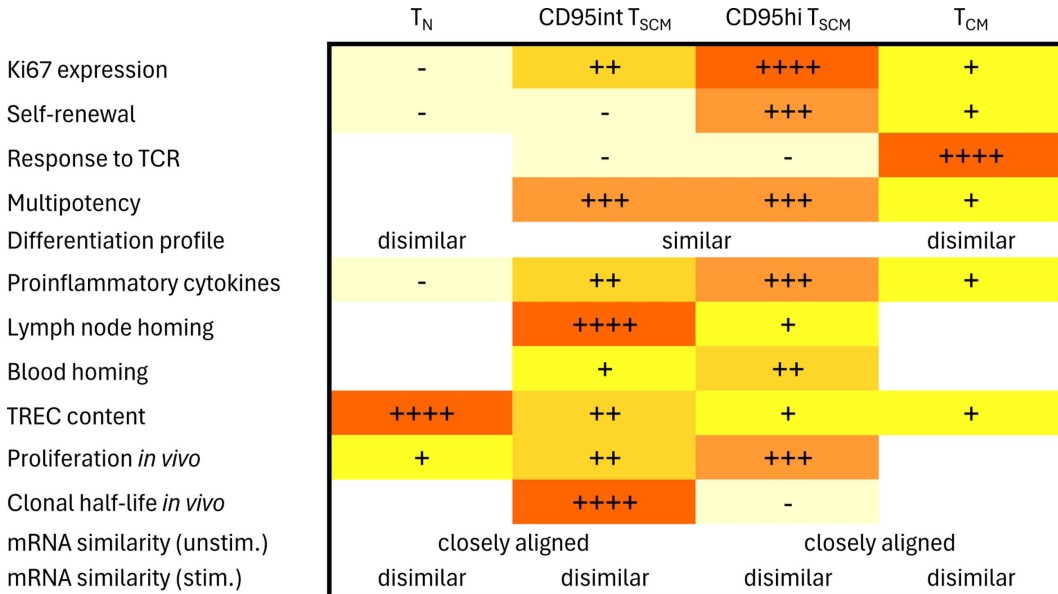

**Fig 12. Summary of $T_{SCM}$ CD95 int and $T_{SCM}$ CD95hi properties.** As well as being distinct from one another in many respects, CD95int $T_{SCM}$ cells and CD95hi $T_{SCM}$ cells are distinct from both $T_N$ and $T_{CM}$ cells. Furthermore, cellular properties do not vary in a monotonic way from $T_N$ to CD95int to CD95hi to $T_{CM}$. A blank cell indicates that combination of property and cell population was not investigated in this study.

An important study from Galletti and colleagues investigated exhaustion in the CD8$^+$ T cell memory compartment [19]. They reported populations of dysfunctional, exhausted cells (defined by coexpression of PD-1 and TIGIT) in the T$_{SCM}$ and T$_{CM}$ populations. They found that once PD1$^+$TIGIT$^+$ cells were excluded from the T$_{SCM}$ and T$_{CM}$ pool then, by RNAseq, there were relatively few differentially expressed genes between the PD1$^-$TIGIT$^-$ T$_{SCM}$ and the PD1$^-$TIGIT$^-$ T$_{CM}$ populations. They suggested that the observed functional differences between T$_{SCM}$ and T$_{CM}$ [4,10,11,28,29] may be attributable to different proportions of exhausted cells in these populations and that once these exhausted cells were excluded, the remaining T$_{SCM}$ and T$_{CM}$ populations could be considered as a single homogeneous population. This interpretation, that dysfunctional status (PD1$^+$TIGIT$^+$) and functional status (PD1$^-$TIGIT$^-$) mark two relatively homogenous populations of T$_{CM}$ and T$_{SCM}$ cells does not appear compatible with our data in which two cell populations with similar levels of PD1$^+$TIGIT$^+$ cells (i.e., CD95int T$_{SCM}$ and T$_{CM}$) differed in many characteristics we measured (Ki67 expression, self-renewal, response to TCR stimulation, production of proinflammatory cytokines, multipotency and TREC content, Fig 12). Moreover, for many features (Ki67 expression, differentiation profile, response to TCR, multipotency, production of proinflammatory cytokines) both CD95hi T$_{SCM}$ and CD95int T$_{SCM}$ differed from T$_{CM}$ in the same direction, again inconsistent with the argument that the only difference between T$_{CM}$ and T$_{SCM}$ is in the proportion of exhausted cells they contain. Finally, we quantified Ki67 expression in T$_{CM}$, T$_{SCM,}$ PD1$^-$TIGIT$^-$ T$_{CM}$ (i.e., excluding exhausted cells) and PD1$^-$TIGIT$^-$ T$_{SCM}$ and found not only that the PD1$^-$TIGIT$^-$ T$_{SCM}$ had significantly higher levels of Ki67 than PD1$^-$TIGIT$^-$ T$_{CM}$ but that exclusion of the exhausted cells did not make the T$_{SCM}$ and T$_{CM}$ populations more similar. Our data support the concept that PD1$^+$TIGIT$^+$ cells can be found in CD95int T$_{SCM}$, CD95hi T$_{SCM}$ and T$_{CM}$ populations but not that the differences between T$_{SCM}$ and T$_{CM}$ observed by others and ourselves can be explained solely by the proportion of these exhausted cells. Our findings are consistent with those of Fuertes Marraco and colleagues [2,4], who in agreement with Galletti and colleagues, found surprisingly little evidence for differentially expressed genes between T$_N$, T$_{SCM}$ and T$_{CM}$ populations (even before exclusion of exhausted cells) but nevertheless found very significant functional differences, particularly in their dynamics: notably the longevity of the YF-vaccine induced population. This indicates that large differences in functionality are not necessarily accompanied by large numbers of differentially expressed genes.

One potential concern regarding our work is the difficulty of sorting cells that are close in flow space. In particular we were concerned that CD95int T$_{SCM}$ might simply be T$_{SCM}$ cells contaminated by T$_N$ and CD95hi T$_{SCM}$ might simply by T$_{SCM}$ cells contaminated by T$_{CM}$ cells. However, this interpretation is not supported by the data. Firstly, for a number of features (self-renewal in response to IL-7/IL-15 and IL-15, differentiation profile in response to IL-2 and IL-15, TNFα, IFNγ and CD107a expression following TCR stimulation), CD95int T$_{SCM}$ are not intermediate between T$_N$ and CD95hi T$_{SCM}$ (Fig 12). Similarly, there are multiple features (Ki67 expression, multipotency, self-renewal in response to both cytokines and TCR stimulation, TNFα, IFNγ and CD107a expression following TCR stimulation) for which CD95hi T$_{SCM}$ are not intermediate between CD95int T$_{SCM}$ and T$_{CM}$ (Fig 12). Secondly, the differentiation profiles of CD95int T$_{SCM}$ and CD95hi T$_{SCM}$ following culture with either IL-15 or IL-2 are not compatible with contamination (Fig C in S1 Text). Another possible concern is that CD95int and CD95hi T$_{SCM}$ are not separate populations but transient states of a single population (indeed it could be imagined that CD95int T$_{SCM}$ are "quiescent T$_{SCM}$" and CD95hi T$_{SCM}$ are "activated T$_{SCM}$"). However, two lines of evidence rule this out. First, we find significant differences in TREC content between CD95int and CD95hi T$_{SCM}$ (Fig 5). If there was frequent interchange between the two subpopulations then these populations would have similar TREC contents. Second, we find that the long-term label enrichment in the CD95int and CD95hi T$_{SCM}$ populations remains distinct for as long as we follow it – establishing that, at least on a time scale of 140 days, there is not frequent interconversion.

Another potential concern is the between-donor variation in cell labelling (Fig 6B). The main reason for this is that the donors varied in the labelling achieved in their saliva (see Panel (A) of Fig K in S1 Text) possibly due to high consumption of "normal" water which would dilute the label in body water or higher levels of exercise; the reduced saliva labelling is adjusted for in calculation of cell kinetics ("Materials and methods") so this will not impact our estimates of proliferation or turnover.

Establishing lineage relationship between cell subpopulations in humans is notoriously difficult. Indeed, 25 yrs since the description of the effector memory and central memory subpopulations [21] there is still no consensus on their lineage topology [28,30,31]. One popular approach is to analyze TCR repertoire sharing. However, the very small size of the $T_{SCM}$ population combined with the highly stochastic nature of TCR repertoire sampling precludes such an approach to studying the relationship between CD95int and CD95hi $T_{SCM}$ cells. Furthermore, observation of shared clones only establishes a common precursor rather than the lineage relationship, i.e., models C, D and E all predict TCR clone sharing between the CD95int and CD95hi $T_{SCM}$ populations (provided sufficient blood volumes could be taken). Our data are most consistent with a linear differentiation model from CD95int $T_{SCM}$ to CD95hi $T_{SCM}$. Perhaps the most compelling evidence is prediction of unseen data (considerably more challenging that fitting seen data) where this linear model outperforms all other lineage relationships considered by a considerable margin (Table L in S2 Text). This is further supported by the *in vitro* differentiation data whereby CD95int $T_{SCM}$ give rise to CD95hi $T_{SCM}$ but CD95hi $T_{SCM}$ tend not to give rise to CD95int $T_{SCM}$ (Fig C in S1 Text). The TREC data and gene expression data also paint a picture consistent with this model. Nevertheless, other lineage topologies cannot be unequivocally ruled out. However, our central conclusion about the extraordinary longevity of clones in the CD95int $T_{SCM}$ compartment and the rapid turnover of the CD95hi compartment is robust to the assumptions regarding the underlying lineage topology.

Our finding is reminiscent of a study in mice in which a cell's division history was estimated in transgenic mice using a synthetic "division-reporter" domain [32]. In this study it was found that murine CD27$^{HI}$KLRG1$^{LO}$ cells (considered a model of human $T_{CM}$) consisted of two subpopulations. One subpopulation (with parallels to the CD95hi $T_{SCM}$ population) had undergone more divisions and had a more effector-like profile while the other was more quiescent and had undergone fewer divisions. Similarly, Zhao and colleagues investigate functional heterogeneity in the human $T_{SCM}$ population and identify two subpopulations on the basis of CD122 expression [18]. Again, there are parallels with our work (CD95hi $T_{SCM}$ cells tend to be CD122hi) and in agreement with our work these cells tended to have increased cytokine production whereas CD122lo cells (enriched in our CD95int population) were transcriptionally more similar to $T_N$ cells. Unfortunately, the study of Zhao and colleagues was limited to *in vitro* flow and RNAseq analysis and did not establish whether CD122hi and CD122lo cells were distinct populations nor their division histories or contribution to the maintenance of memory.

Another interesting parallel is with hematopoiesis. It has been shown that the cells at the apex of the hematopoietic stem cell hierarchy – the hematopoietic stem cells (HSCs) – divide very infrequently with the frequency of division increasing for more mature stem cell populations such as long-term HSCs and multipotent progenitors which also show improved self-renewal [33]. $T_{SCM}$ cells appear to follow the same schema, with the most immature cells, the CD95int $T_{SCM}$ subpopulation, dividing very infrequently (on average once every 3 yrs) whereas the more mature CD95hi $T_{SCM}$ subpopulation divides more frequently (approximately once a year) and have significantly higher levels of self-renewal.

There is considerable interest in $T_{SCM}$ cells in the clinical and translational space. Response to the adoptive transfer of T cells for treatment of metastatic melanoma patients has been linked to the presence of $T_{SCM}$ cells in the infused treatment product; and the beneficial effect seems to be tightly linked with these clones ability to survive long term [34]. Similarly stem-like T cells appear to be the population responsible for a successful response to immune-checkpoint blockade (ICB). In a study of responder and non-responder patients receiving ICB, the ratio of stem-like TCF7hi: TCF7lo cells infiltrating the tumor was predictive of responsiveness [35]. $T_{SCM}$ cells are also being explored as the basis for chimeric antigen receptor (CAR)-T cells [36]. All these approaches leverage the unique longevity and self-renewal properties of $T_{SCM}$ cells. These same key dynamic properties can have deleterious effects in other contexts. $T_{SCM}$ cells have been linked to the development of adult T cell leukemia, to the survival of CD4$^+$ cells latently infected with HIV-1 as well as to the development of type I diabetes [37–39]. Therapeutic targeting of $T_{SCM}$ cells – either to enhance the efficacy of cancer immunotherapies, to enhance the longevity of the vaccine-induced response or to block $T_{SCM}$-driven diseases – is predicated on an understanding of the cells in question. Our finding that there is marked heterogeneity in the unique dynamic properties of this cell population is crucial information.

CD95 (FAS or APO-1) is a member of the tumor necrosis factor (TNF) receptor family. The FAS-FASL pathway plays a critical role in the control of the immune system; it maintains homeostasis by killing unwanted cells and is also important for activation induced cell death of T cells during a chronic immune response [40]. In common with other members of the TNF family, CD95 has a dual role. It can act as a death inducing receptor leading to apoptosis, while also mediating other functions including proliferation and tissue regeneration [41]. It has been demonstrated that T cell proliferation induced by sub-optimal TCR engagement is enhanced when CD95 is bound [42]; while deletion of CD95 in T cells leads to lymphopenia in mice [43]. In activated T cells, CD95 acts as a dual-function receptor: exposure to low levels of CD95L leads to survival but exposure to high concentrations of CD95L leads to apoptosis [44]. Together, these data demonstrate that CD95 expression is involved in T cell proliferation, survival, and activation.

The data we present here are most consistent with a picture in which $T_N$ cells meet cognate antigen, undergo proliferation and a proportion differentiate into the CD95int $T_{SCM}$ pool. Cells in this CD95int $T_{SCM}$ population proliferate slowly (median 0.0008 per day, Table 1 in main text), faster than $T_N$ cells (median 0.0003 per day) but still slower than other memory T cell subpopulation [14]. Clones in this compartment are extraordinarily long-lived (order of decades); indeed, they are longer lived than clones of any memory subpopulation defined to date [14]. A finding that was strikingly confirmed by the YFV tetramer data. This population is predominantly found in the lymph node. In response to TCR stimulation these cells are more reactive than $T_N$ or $T_{CM}$ (but considerably less than $T_{EM}$). These cells have a transcriptional profile similar to $T_N$ cells and their frequency decreases with age. In response to homeostatic cytokines CD95int $T_{SCM}$ cells have limited self-renewal properties. CD95int $T_{SCM}$ clones are long-lived not because they self-renew but because they are largely quiescent. In contrast, CD95hi $T_{SCM}$ cells are short-lived both at the cellular and at the clonal level. When they divide under homeostatic conditions they retain their phenotype (i.e., they self-renew) but in response to TCR stimulation they are highly multipotent and differentiate into all downstream memory subsets ($T_{CM}$, $T_{EM}$, $T_{EMRA}$). CD95hi $T_{SCM}$ cells are found in higher frequencies in the blood than lymph node, accumulate with age, have a much lower TREC content than CD95int $T_{SCM}$ and have a transcriptional profile that is more similar to $T_{CM}$. *In vivo* they proliferate rapidly ($p$ = 0.002 per day) but have short lived clones (life span of the order of months) and are sustained by differentiation of cells from the CD95int $T_{SCM}$ pool. Long-term *in vivo* labelling, establishes that CD95int $T_{SCM}$ and CD95hi $T_{SCM}$ are two distinct populations not transient states of the same cell population.

Stem cells are considered to be cells with three main properties: self-renewal, clonal longevity and multipotency. While both $T_{SCM}$ subpopulations exhibit considerable multipotency, no single population of $T_{SCM}$ cells has both the dynamic properties of self-renewal and clonal longevity. Instead, the "stemness" of the $T_{SCM}$ population is generated by the complementary properties of two distinct subpopulations: CD95int $T_{SCM}$ which have the property of clonal longevity and CD95hi $T_{SCM}$ which have the properties of expansion and self-renewal. We suggest that together, these two populations function as a stem cell population.

## Materials and methods

### Ethics

All study procedures were conducted according to the principles of the Declaration of Helsinki and all participants gave written informed consent following protocols approved by the relevant ethics committee (NRES London 13/LO/0022, ICREC 21IC7146, 15/SC/0089/2, Fulham REC 18/LO/2196). Some of the human samples used in this research project were obtained from the Imperial College Healthcare Tissue and Biobank (ICHTB). ICHTB is supported by the National Institute for Health Research (NIHR) Biomedical Research Centre based at Imperial College Healthcare NHS Trust and Imperial College London. ICHTB is approved by Wales REC3 to release human material for research (22/WA/0214).

## Study subjects

We studied a total of 106 healthy volunteers.

Cohort 1.  *Ex vivo* $T_{SCM}$ analysis and qPCR ($N = 25$). Donors were aged 18–60 yrs and healthy at the time of donation with no symptomatic or diagnosed infections, allergies or autoimmune diseases.

Cohort 2.  *Ex vivo* $T_{SCM}$ analysis: replication ($N = 8$). Donors were aged 18–60 yrs and healthy at the time of donation with no symptomatic or diagnosed infections, allergies or autoimmune diseases.

Cohort 3.  Lymph node fine needle aspirate ($N = 7$). Lymph node cells were taken from left and/or right lymph nodes from 7 healthy, vaccinated HIV-negative volunteers aged 20–44 yrs enrolled into the EAVI2020_01 study [22].

Cohort 4.  RNAseq, TREC and phenotype analysis ($N = 21$). Donors were aged 27–65 yrs and healthy at the time of donation with no symptomatic or diagnosed infections, allergies or autoimmune diseases.

Cohort 5.  *In vivo* stable isotope labelling ($N = 3$). Donors were aged 38 yrs (DW19), 42 yrs (DW20) and 56 yrs (DW25) and healthy at the time of donation with no symptomatic or diagnosed infections, allergies or autoimmune diseases.

Cohort 6.  Telomere analysis ($N = 5$). Donors were aged 29−83 yrs. All donors were CMV-seropositive and HIV-1-seronegative. This cohort has previously been described [12].

Cohort 7.  Yellow fever vaccination cohort ($N = 37$). Donors were *HLA-A*02*+ and had received the yellow fever vaccination YF-17D at different timepoints in the past (range 0.27–35.02 yrs). This cohort has previously been described [2,12,29].

## Flow cytometry: Analysis

Peripheral blood mononuclear cells were stained with a cocktail of fluorescently labelled antibodies (see Table A in S2 Text for clone, supplier and dilution). For intracellular analysis, cells were fixed/permeabilized using the FoxP3 Transcription Factor Staining Buffer Set (eBioscience, Thermo Fisher Scientific, Massachusetts, US) as per manufacturer's instructions. Cells were then re-suspended in FACS buffer for acquisition. Proliferation was assessed by measuring the progressive dilution of CellTraceViolet (CTV) dye (Thermo Fisher Scientific). Samples were acquired using a BD LSRFortessa (BD Biosciences, New Jersey, US). Compensation was performed using the UltraComp eBeads (Invitrogen, ThermoFisher Scientific) as single-stained controls. Individual cell populations were gated using fluorescence minus one (FMO) controls. Data were quantitated using FlowJo (v9.0). The separation between CD95int $T_{SCM}$ and CD95hi $T_{SCM}$ was defined based on backgating of Ki67 with gates initially set to provide maximum discrimination between Ki67+ and Ki67− cells. Henceforth, the same gating was used for all individuals except for some cases where the CD8+ subpopulations were visibly shifted and so the CD95hi/int gate was moved based on the location of subpopulations in that individual. On changing to a new machine (e.g., sorter) then this process was repeated to obtain the corresponding cutoff for the new machine.

## Flow cytometry: Sorting

T cell subsets were sorted to purity (>80%) using a FACSAria III (BD Biosciences). CD8+ T cells were sorted into the following subpopulations based on the expression of canonical differentiation markers: $T_N$ (naïve: CD45RA+CCR7+CD95−), CD95int $T_{SCM}$ (CD95 intermediate T stem cell like-memory: CD45RA+CCR7+CD95INT), CD95hi $T_{SCM}$ (CD95 high T stem cell like-memory: CD45RA+CCR7+CD95HI), $T_{CM}$ (central memory: CD45RA−CCR7+), $T_{EM}$ (effector memory: CD45RA−CCR7−) and $T_{EMRA}$ (CD45RA-expressing effector memory: CD45RA+CCR7−). See Fig B in S1 Text for representative gating.

**PLOS Biology**

### *In vitro* culture

FACS-purified CD8⁺ T cell subsets were cultured in RPMI 1640 medium (Thermo Fisher Scientific) supplemented with 10% FBS (heat inactivated, Gibco), 1% penicillin-streptomycin and 2 mM L-glutamine (Sigma–Aldrich, Missouri, US). To evaluate self-renewal under homeostatic conditions, cells were stimulated for 7 d either with recombinant human interleukin 7 (IL-7) and interleukin (IL-15) (both at 25 ng/mL, Miltenyi Biotec, Bergisch Gladbach, Germany) or 25 ng/mL IL-15 or 6.1 ng of IL-2. The cytokines were renewed on day 3. To evaluate differentiation, cells were stimulated for 7 days with purified anti-human plate bound (pb) αCD3 and soluble αCD28 (2 µg/ml and 1 µg/ml, respectively, BD Biosciences) in the presence of IL-7 and IL-15. The plates were pre-coated with pb αCD3 for 2 h at 37 °C. To measure cytokine production and cytotoxicity, cells were stimulated for 12 h with pb αCD3 (2 µg/mL) and soluble αCD28 (1 µg/ml). GolgiStop (1×) and GolgiPlug (1×) (both BD Biosciences) were added during the last 10 h of stimulation. Cells were then removed from the stimulation and stained for surface and intracellular markers as described above. For CD107 staining, the antibody was added at the beginning of the stimulation and cells were not re-stained during the surface staining step. Following Gattinoni and colleagues [1], the multipotency index for a cell subpopulation with index $j$ was calculated using the following formula:

$$MI_j = -\sum_{\substack{i=1 \\ i \neq j}}^{n} p_i \log_{10}(p_i)$$

(1)

Where $p_i$ is the proportion of differentiated cells at day 7 that have phenotype $i$ and $n$ is the number of phenotypes considered ($n = 6$).

### Cell trace violet staining

Proliferation was calculated by measuring the progressive dilution of CellTraceViolet (CTV) dye (Thermo Fisher Scientific). CTV was diluted at 5 µM in warm PBS and cells were stained for 20 min at room temperature prior to surface staining. FACS buffer was added at the end of the incubation for 5 min to inactivate the dye.

### Gene expression analysis: qPCR

The RNAeasy Micro Kit (Qiagen) was used for RNA isolation as per manufacturer's instructions. RNA isolation was followed by cDNA conversion on the same day using the TaqMan Reverse transcription reagents (Thermo Fisher Scientific) and a SimpliAmp Thermo Cycler (Applied Biosystems, Massachusetts, US). The cDNA was stored at −20 °C until needed. The transcript levels were assessed via qPCR using TaqMan Fast Advanced Master Mix and TaqMan primer probes (all Thermo Fisher Scientific) as per manufacturer's protocol. See Table B in S2 Text for a list of genes studied and the corresponding probes. qPCR Assays were run on a QuantStudio 7 Flex Real-Time PCR System (Thermo Fisher Scientific) for 45 cycles and gene expression relative to housekeeping gene 18S rRNA was calculated as $2^{-\Delta CT}$; where ΔCT is the difference between the cycle threshold (CT) value for the gene of interest and the CT for the housekeeping gene. All subjects from Cohort 1.

### Gene expression analysis: Bulk RNAseq

CD8⁺ $T_N$, CD95int $T_{SCM}$, CD95hi $T_{SCM}$, $T_{CM}$, $T_{EM}$ and $T_{EMRA}$ populations were sorted from PBMC of 12 healthy donors. Cells were cultured in RPMI 1640 medium (Thermo Fisher Scientific) supplemented with 10% FBS (heat-inactivated, Gibco), 1% penicillin-streptomycin and 2 mM L-glutamine (Sigma–Aldrich, Missouri, US) for 12 h ("unstimulated") or cultured as above with the addition of plate-bound αCD3 (2 µg/mL) and soluble αCD28 (1 µg/ml) at 37 °C ("stimulated"). See Table C in S2 Text for experimental design. RNA extraction was performed utilizing the RNeasy Micro Plus Kit (Qiagen) following

the manufacturer's instructions. The quality and quantity of the isolated RNA were assessed using the RNAseq experiment Agilent RNA 6,000 Pico Kit (Agilent Technologies) following the manufacturer's protocols. Subsequently, the RNA samples were preserved at −80 °C until they were ready for further processing. RNA sequencing was conducted at the Oxford Genomics Centre, where polyAenriched, strand-specific libraries were generated using the NEBNext Ultra II Directional RNA Library Prep Kits (Illumina). All subjects from Cohort 4.

**Generation of gene counts from raw reads.** FASTQ files were processed with nf-core/rnaseq bioinformatics pipeline [45] using the STAR-RSEM alignment/quantification methods. Briefly, reads are aligned to the reference genome (GRCh38) using STAR v2.7.10a [46] and gene expression is quantified using RSEM v1.3.1 to deal with ambiguously mapped reads [47]. Mapping statistics and quality control metrics from FastQC and MultiQC reports indicated high data quality for all samples but for 2, where the number of mapping reads was extremely low. These two samples were discarded. Downstream data analysis was performed in R v4.2.1.

**Batch correction of gene counts.** The 80 gene count matrices generated with RSEM were read into R with the tximport() function from tximportData package [48]. Gene features that did not have at least 10 counts in at least 3 samples (the size of the smallest biological subgroup, i.e., stimulated $T_{cm}$ samples) were discarded. Since samples were obtained from different donors, processed at different times and RNA was extracted in 3 separate batches, batch correction was performed to remove batch effects in an unsupervised manner. Briefly, gene counts were first normalized using the median of ratios method to account for differences in library size and RNA composition between samples [49]. Then, normalized counts were transformed using variance stabilizing transformation with the vst() function from DESeq2 package so differences between samples are not overrepresented by highly expressed genes. To remove batch effects, the sva() function from the SVA package function is applied to the vst gene counts and with *model ~ population + condition + population:condition* (where population: $T_{naive}$, $T_{scm}$ CD95$^{hi}$, $T_{scm}$ CD95$^{int}$, $T_{cm}$, $T_{em}$ and $T_{emra}$ and condition: stimulated and unstimulated). sva() identified 26 surrogate variables that were regressed out from vst counts using the fsva() function. The batch corrected matrix was used for exploratory analysis and QC to visualize the relationship between the samples. Specifically, three different approaches were used for exploratory analysis: principal component analysis (PCA) with prcomp(), hierarchical clustering with hclust() and Euclidean distances between samples using dist(), all functions from the stats package. Four samples had a highly skewed vst count distribution (D11_Hi_S, D3_Int_S, D13_Int_U and D13_Hi_U) and were excluded from downstream DE analysis.

**Differential expression analysis with Wilcoxon Rank Sum test.** The wilcox.exact() function from exactRankTests package was used for statistical testing. Briefly, the DESeqDataSet was first filtered to keep unstimulated samples only. Counts from this subset of unstimulated samples were vst transformed after normalization with respect to library size. Surrogate variables were then re-calculated with sva() on the vst transformed counts. The estimated surrogate variables were regressed out from the vst counts. Unstimulated $T_{scm}$ CD95hi samples were then compared to unstimulated $T_{scm}$ CD95int samples with Wilcoxon rank sum test. Only genes that had at least 10 counts in at least 6 samples and had GO annotations were tested. *P*-values were FDR-adjusted for multiple comparisons using the p.adjust() function. Genes with adjusted *p*-value < 0.05 were considered differentially expressed. The same steps were repeated to compare stimulated $T_{scm}$ CD95hi samples to stimulated $T_{scm}$ CD95int samples.

**Enrichment analysis.** Pre-ranked gene set enrichment analysis was performed with the fgsea() function from fgsea package [50] and with the GSEA() function from DOSE package [51]. Gene sets tested for enrichment were downloaded from MSigDB [52,53] and only included immune-related pathways (C7: immunologic signature gene sets, [54]). Gene-concept networks to visualize enrichment results were generated with enrichplot package [55].

**Heatmaps.** Heatmaps showing the top 70 DEG were generated using the R packages pheatmap and ComplexHeatmap. Note that SVA corrected counts were scaled to obtain z-scores by row, i.e., by gene. Both samples (columns) and genes (rows) are clustered based on Euclidean distance and complete method and using z-scores.

## Lymph node cell collection

Ultrasound guided fine needle aspiration was used to collected axillary lymph node samples from volunteers participating in an experimental vaccine immunogen study using an HIV Env Mosaic immunogen (Cohort 3) [22]. Cells were taken either from the left or right lymphoid tissues, i.e., contralateral and ipsilateral to the site of injection (and in some cases both); matched blood samples from the same individuals at the same time point were also collected. Cells were collected between 7 days and 56 days post-first immunization but before the subsequent boost, a time point at which antigen-specific IgG was undetectable [22]. In cases where data were available from both left and right lymph node we first ascertained that the measurements were independent (no significant correlation) and then pooled the data.

## T cell receptor excision circle (TREC) quantification

To quantify TRECs from sorted memory subpopulation we used our in-house droplet digital PCR protocol directly from cell lysate [23]. Cell lysate was made using lysis reagents from the SuperScript IV CellsDirect cDNA Synthesis Kit (ThermoFisher). Cell lysate (30 µl approximately) was prepared according to manufacturer's instructions, omitting the DNA degradation step. The prepared cell lysate was then heated on a thermocycler at 65 °C for 1 min, 96 °C for 2 min, 65 °C for 4 min, 96 °C for 1 min, 65 °C for 1 min and 96 °C for 30 s. Tubes were spun down for 1 min and cooled at room temperature. From the prepared cell lysate, 7.8 µl were added to the reaction mixture containing 11 µl 2× ddPCR Supermix for probes (no dUTP) (Bio-Rad, Pleasanton, CA, USA), 0.55 µl of TRECs primers (36 µM stock) and probes (10 µM) each, 1.1 µl of RPP30 copy number detection (Bio-Rad), 1 µl of HindIII-HF restriction enzyme (NEB, UK). The sequences of primers and probe used for TRECs are: Forward primer 5′-CAC ATC CCT TTC AAC CAT GCT-3′; Reverse primer 5′-GCC AGC TGC AGG GTT TAG G-3′, Probe Sequence: 5′-ACA CCT CTG GTT TTT GTA AAG GTG CCC ACT-3′ with FAM and BQ-1 quencher. For RPP30, we used the primer probe mix from BioRad. Four replicates were made. During data analysis, these replicates were summed to give a total of 80,000 droplets/sample. To make the droplets, 20 µl reaction mixture was loaded into sample loading wells and 70 µl of droplet oil was loaded in the droplet oil wells in a droplet cartridge (Bio-Rad, USA). The cartridge was covered with a gasket and placed in the droplet generator (Bio-Rad #186-3002). An amount of 40 µl of droplets were transferred to a 96-well plate (Bio-Rad, USA). The plate was put on the thermocycler (Bio-Rad, USA) under the following conditions: 10 min hold at 95 °C, 45 cycles of 95 °C for 15 s then 59 °C for 60 s. Amplified droplets were then read in the QX200 Droplet Reader (Bio-Rad, USA). The data were analyzed in QuantaSoft analysis software version 1.7 (Bio-Rad).

## Stable isotope labelling *in vivo*

We have previously described the labelling protocol in detail [13,56]. Briefly, participants (Cohort 5) were given oral doses of 70% deuterated water ($^2H_2O$) over a 7-week period (100 ml twice daily for one day, then 50 ml twice daily for 48 days). Saliva samples were collected for evaluation of body water labelling. Peripheral blood was collected at successive time-points during and after labelling and PBMC separated by Ficoll gradient centrifugation. Monocytes for normalization, as a cell population expected to approach fully-labelled status during the labelling phase, were sorted from an aliquot of PBMC by CD14 magnetic bead column positive selection (MACS, Miltenyi Biotech, Bisley, UK). PBMC were sorted using a BD FACSAria III flow cytometer into CD8$^+$ T$_N$, CD8$^+$ CD95int T$_{SCM}$ and CD8$^+$ CD95hi T$_{SCM}$. Deuterium enrichment in the DNA of monocytes and sorted T cell subpopulations was measured by gas chromatography/mass spectrometry of the pentafluorobenzyl derivative using a Shimadzu 2030–QP2020NX with SHIM35MS column (Shimadzu, Milton Keynes, UK) as previously described [24,57].

## Single-chromosome telomere length analysis

These data have previously been reported [12]. Briefly, DNA from CD8$^+$ T$_N$ and total CD8$^+$ T$_{SCM}$ cells (not separated into CD95hi and CD95int subpopulations) was extracted, and single-telomere length analysis was carried out at the XpYp telomere [58].

## Yellow fever vaccine data

Published data were acquired from a cross-sectional study of 37 healthy adults who received a single dose of the yellow fever vaccine YF-17D (Cohort 7) [2]. Time since vaccination ranged from 3 mo to 35 yrs. Four subjects who received multiple YF vaccinations were excluded from the analysis. Total CD8$^+$CCR7$^+$CD45RA$^+$ T$_{SCM}$ cells (not separated into CD95hi and CD95int subpopulations) specific for the HLA-A*02-restricted YFV NS4b214-222 epitope were quantified by flow cytometry with antibody and tetramer staining.

## Modelling the data: (1) Stable isotope labelling data

There are three steps in the modelling to estimate the proliferation and disappearance rate of the T cell subsets. First, we quantify the availability of label in body water by measuring the fraction of heavy water in the saliva. We describe this label availability with an empirical function $S(t)$ with three parts (equation 2) to reflect the three part protocol (full dose 1 day, half dose 48 days, delabel).

$$S(t) = \begin{cases} f(1 - e^{-\delta t}) & t < 1,1 \\ {}^1\!/_2 f(1 - e^{-\delta(t-1)}) + f(1 - e^{-\delta})e^{-\delta(t-1)} & 1 < t \leq 49 \\ \left[ {}^1\!/_2 f(1 - e^{-\delta(49-1)}) + f(1 - e^{-\delta 1})e^{-\delta(49-1)} \right] e^{-\delta(t-49)} & t > 49 \end{cases}$$

(2)

Next, for each individual we model the fraction of label in a rapidly turning over population (monocytes) in order to estimate the amplification factor $b_w$ (also referred to as $c$ [59]); this is a factor that reflects the increase in M + 1 when a cell divides given enrichment $S(t)$ it scales between label enrichment in newly synthesized DNA and precursor availability in body water [14]. We describe the label enrichment in DNA of monocytes using a mechanistic model previously proposed [60,61]:

$$\dot{L}_M = p_m b_w S(t) - r_1 L_M$$
$$\dot{L}_B = r_1 L_M(t - \Delta) \frac{M}{B} - r_2 L_B$$

(3)

Where $L_M$ is the fraction of label in bone marrow monocyte precursors, $L_B$ is the fraction of label in blood monocytes (the observable), $p_m$ is the proliferation rate of precursors, $r_1$ is the rate of exit from the mitotic pool in bone marrow, $\Delta$ is the time spent in the post-mitotic pool in bone marrow, $M/B$ is the ratio of the number of monocytes in the bone marrow to the number of monocytes in the blood, $S(t)$ is the saliva enrichment estimated for that individual in step 1 and $b_w$ is the amplification factor of interest. A schematic of this model is provided in Fig J in S1 Text. We use equilibrium constraints to eliminate $p_m$ and $r_1$. $M/B$ and $\Delta$ were fixed at estimates of 2.6 and 1.6 days, respectively [60,61] (however, we showed that the estimates of $b_w$ are independent of these values; this follows because $b_w$ depends only on the plateau enrichment in blood monocytes). There were therefore two free parameters ($b_w$ and $r_2$) which were estimated by fitting the model to the data. Finally, we use the parameter estimates ($f$, $\delta$, $b_w$) from steps 1 and 2 to describe the label enrichment in T cells.

Five different models of T cell dynamics were developed. Each model encapsulates different assumptions about the relationship between the three T cell populations of interest (T$_N$, CD95int, CD95hi) and their corresponding dynamics. The aim was to see which model was most consistent with the data and to estimate the corresponding parameters. The models are described below and illustrated in Fig 7.

**Model A. Independent homogeneous model.** Each cell population (T$_N$, CD95int, CD95hi) is independent of the others and kinetically homogeneous (i.e., they do not contain subpopulations with different kinetics). Cell numbers in this model are described by the following equations:

$$\dot{T}_N = (p_N - d_N)T_N$$
$$\dot{T}_1 = (p_1 - d_1)T_1$$
$$\dot{T}_2 = (p_2 - d_2)T_2 \qquad (4)$$

And label is described by the following:

$$\dot{L}_N = p_N b_w S(t) - d_N L_N$$
$$\dot{L}_1 = p_1 b_w S(t) - d_1 L_1$$
$$\dot{L}_2 = p_2 b_w S(t) - d_2 L_2 \qquad (5)$$

Where $T_N$, $T_1$, $T_2$ are the sizes of the $T_N$, CD95int and CD95hi populations, respectively, $L_N$, $L_1$, $L_2$ is the fraction of label in the $T_N$, CD95int and CD95hi populations, respectively; $p_N$, $p_1$, $p_2$ is the proliferation rate of the $T_N$, CD95int and CD95hi populations, respectively; $d_N$, $d_1$, $d_2$ the disappearance rates of the $T_N$, CD95int and CD95hi populations, respectively; $b_w$ is the amplification factor and $S(t)$ the label in the saliva (equation 2). We use equilibrium constraints ($T_N$, $T_1$ and $T_2$ assumed to be independently in equilibrium in (equation 4)) to eliminate $d_N$, $d_1$ and $d_2$.

**Model B. Independent heterogeneous model.** Each cell population ($T_N$, CD95int and CD95hi) is independent of the others and allowed to be kinetically heterogeneous (i.e., they can contain subpopulations with different kinetics). Cell numbers in this model are described by the following equations:

$$\dot{T}_N = (p_N - d_N)T_N$$
$$\dot{T}_1 = (p_1 - d_1)T_1$$
$$\dot{T}_2 = (p_2 - d_2)T_2 \qquad (6)$$

Label in this model is described by the following equations:

$$\dot{L}_N = p_N b_w S(t) - d_N^* L_N$$
$$\dot{L}_1 = p_1 b_w S(t) - d_1^* L_1$$
$$\dot{L}_2 = p_2 b_w S(t) - d_2^* L_2 \qquad (7)$$

Where $T_N$, $T_1$, $T_2$ are the sizes of the $T_N$, CD95int and CD95hi populations, respectively; $L_N$, $L_1$, $L_2$ is the fraction of label in the $T_N$, CD95int and CD95hi populations, respectively; $p_N$, $p_1$, $p_2$ is the proliferation rate of the $T_N$, CD95int and CD95hi populations, respectively; $d_N$, $d_1$, $d_2$ is the disappearance rate of the $T_N$, CD95int and CD95hi populations, respectively; $d_N^*$, $d_1^*$, $d_2^*$ the disappearance rates of labelled cells in the $T_N$, CD95int and CD95hi populations, respectively; $b_w$ is the amplification factor and $S(t)$ the label in the saliva. In general, due to heterogeneity [62], the disappearance rate of labelled cells is not equal to the disappearance rate of the whole population ($d_N \neq d_N^*$ etc.) and so equilibrium constraints are not helpful here (since $d_N$, $d_1$, $d_2$ do not appear in equation 7).

**Model C. Forked differentiation model.** Upon meeting cognate antigen, $T_N$ cells undergo a clonal burst of size $k$ (i.e., undergo $k$ divisions and produce $2^k$ cells). A fraction $f$ differentiate into CD95int, the remainder $(1 - f)$ differentiate into CD95hi. Cell numbers in this model are described by

$$\dot{T}_N = (p_N - d_N - \Delta_N)T_N$$
$$\dot{T}_1 = (p_1 - d_1)T_1 + \Delta_N f 2^k T_N$$
$$\dot{T}_2 = (p_2 - d_2)T_2 + \Delta_N (1 - f) 2^k T_N \qquad (8)$$

And label in this model is described by:

$$\dot{L}_N = p_N b_w S(t) - (d_N + \Delta_N)L_N$$
$$\dot{L}_1 = p_1 b_w S(t) - d_1 L_1 + \Delta_N f(2^k - 1)b_w S(t)/R_1 + \Delta_N f L_N/R_1$$
$$\dot{L}_2 = p_2 b_w S(t) - d_2 L_2 + \Delta_N (1-f)(2^k - 1)b_w S(t)/R_2 + \Delta_N (1-f)L_N/R_2 \tag{9}$$

Where $T_N$, $T_1$, $T_2$ are the sizes of the $T_N$, CD95int and CD95hi populations, respectively; $L_N$, $L_1$, $L_2$ is the fraction of label in the $T_N$, CD95int and CD95hi populations, respectively; $p_N$, $p_1$, $p_2$ is the proliferation rate of the $T_N$, CD95int and CD95hi populations, respectively; $d_N$, $d_1$, $d_2$ the disappearance rates of the $T_N$, CD95int and CD95hi populations, respectively; $\Delta_N$ the fraction of $T_N$ undergoing differentiation per day, $k$ the size of the clonal burst, $f$ the fraction of expanded cells that differentiate into the CD95int population, $R_1$ the ratio of CD95int to $T_N$ cells, $R_2$ the ratio of CD95hi to $T_N$ cells, $b_w$ is the amplification factor and $S(t)$ the label in the saliva. We use equilibrium constraints to eliminate $d_N$, $d_1$ and $d_2$. For the case when we fit only the labelling data we reparameterized this model to improve parameter identifiability. We used

$$\Pi_1 = p_1 + \Delta_N f(2^k - 1)/R_1$$
$$\Pi_2 = p_2 + \Delta_N (1-f)(2^k - 1)/R_2 \tag{10}$$

This allows us to eliminate $p_1$, $p_2$ and $k$ leaving us with 5 free parameters for the label only fits ($p_N$, $\Pi_1$, $\Pi_2$, $f$, $\Delta_N$). For the case when we are fitting all the data (labelling, telomere, YFV), then the way in which the parameters appear in different combinations in the models of the different types of data precludes this reparameterization and we use the original parameterization when fitting all the data.

**Model D. Linear differentiation model (CD95int first).** Upon meeting cognate antigen, $T_N$ cells undergo a clonal burst of size $k$ (producing $2^k$ cells), these differentiate first into CD95int and subsequently into CD95hi. Cell numbers are described by the following:

$$\dot{T}_N = (p_N - d_N - \Delta_N)T_N$$
$$\dot{T}_1 = \Delta_N T_N 2^k + T_1 p_1 - (d_1 + \Delta_1)T_1$$
$$\dot{T}_2 = \Delta_1 T_1 + (p_2 - d_2)T_2 \tag{11}$$

Label in this model is described by the following equations:

$$\dot{L}_N = p_N b_w S(t) - (d_N + \Delta_N)L_N$$
$$\dot{L}_1 = p_1 b_w S(t) - (d_1 + \Delta_1)L_1 + \Delta_N (2^k - 1)b_w S(t)/R_1 + \Delta_N L_N/R_1$$
$$\dot{L}_2 = p_2 b_w S(t) - d_2 L_2 + \Delta_N (2^k - 1)b_w S(t)/R_2 + \Delta_N L_N/R_2 \tag{12}$$

Where $T_N$, $T_1$, $T_2$ are the sizes of the $T_N$, CD95int and CD95hi populations, respectively; $L_N$, $L_1$, $L_2$ is the fraction of label in the $T_N$, CD95int and CD95hi populations, respectively; $p_N$, $p_1$, $p_2$ is the proliferation rate of the $T_N$, CD95int and CD95hi populations, respectively; $d_N$, $d_1$, $d_2$ the disappearance rates of the $T_N$, CD95int and CD95hi populations, respectively; $\Delta_N$ the fraction of $T_N$ undergoing differentiation per day, $k$ the size of the clonal burst, $\Delta_1$ the rate of differentiation from CD95int to CD95hi, $R_1$ the ratio of CD95int to $T_N$ cells, $R_2$ the ratio of CD95hi to $T_N$ cells, $b_w$ is the amplification factor and $S(t)$ the label in the saliva. We use equilibrium constraints to eliminate $d_N$, $d_1$ and $d_2$.

**Model E. Linear differentiation model (CD95hi first).** Upon meeting cognate antigen, $T_N$ cells undergo a clonal burst of size $k$ (producing $2^k$ cells), these differentiate first into CD95hi and subsequently into CD95int. Cell numbers in this model are described by the following equations

$$\dot{T}_N = (p_N - d_N - \Delta_N)T_N$$
$$\dot{T}_2 = \Delta_N T_N 2^k + (p_2 - d_2 - \Delta_1)T_2$$
$$\dot{T}_1 = \Delta_1 T_2 + (p_1 - d_1)T_1$$
(13)

Label in this model is described by the following equations:

$$\dot{L}_N = p_N b_w S(t) - (d_N + \Delta_N)L_N$$
$$\dot{L}_2 = p_2 b_w S(t) - (d_2 + \Delta_1)L_2 + \Delta_N(2^k - 1)b_w S(t)/R_2 + \Delta_N L_N/R_2$$
$$\dot{L}_1 = p_1 b_w S(t) - d_1 L_1 + \Delta_N(2^k - 1)b_w S(t)/R_1 + \Delta_N L_N/R_1$$
(14)

Where $T_N$, $T_1$, $T_2$ are the sizes of the $T_N$, CD95int and CD95hi populations, respectively; $L_N$, $L_1$, $L_2$ is the fraction of label in the $T_N$, CD95int and CD95hi populations, respectively; $p_N$, $p_1$, $p_2$ is the proliferation rate of the $T_N$, CD95int and CD95hi populations, respectively; $d_N$, $d_1$, $d_2$ is the disappearance rate of the $T_N$, CD95int and CD95hi populations, respectively; $\Delta_N$ the fraction of $T_N$ undergoing differentiation per day, $k$ the size of the clonal burst, $\Delta_1$ the rate of differentiation from CD95hi to CD95int, $R_1$ the ratio of CD95int to $T_N$ cells, $R_2$ the ratio of CD95hi to $T_N$ cells $b_w$ is the amplification factor and $S(t)$ the label in the saliva. We use equilibrium constraints to eliminate $d_N$, $d_1$ and $d_2$.

### Modelling the data: (2) Telomere data

Equations predicting the difference in mean telomere length between $T_N$ and total $T_{SCM}$ ($\theta$) were derived following [12,63] for each of the 5 models.

**Model A. Independent homogeneous model and Model B. Independent heterogeneous model.**

$$\theta = \left( \frac{R_1}{R_1 + R_2} 2p_1 + \frac{R_2}{R_1 + R_2} 2p_2 - 2p_N \right) \delta t$$
(15)

Where $p_N$, $p_1$, $p_2$ is the proliferation rate of the $T_N$, CD95int and CD95hi populations, respectively; $R_1$ the ratio of CD95int to $T_N$ cells, $R_2$ the ratio of CD95hi to $T_N$ cells, $\delta$ is the number of bases lost per division (assumed to be 50 bp) and $t$ is the age of the individual (in days).

**Model C. Forked differentiation model.**

$$\dot{\theta} = \left( \frac{R_1}{R_1 + R_2} \dot{\mu}_1 + \frac{R_2}{R_1 + R_2} \dot{\mu}_2 - \dot{\mu}_N \right) \delta$$
where
$$\dot{\mu}_1 = 2p_1 + 2^k \Delta_N f(\mu_N + k - \mu_1)/R_1$$
$$\dot{\mu}_2 = 2p_2 + 2^k \Delta_N (1-f)(\mu_N + k - \mu_2)/R_2$$
$$\dot{\mu}_N = 2p_N$$
(16)

Where $\mu_N$, $\mu_1$, $\mu_2$ is the telomere length of $T_N$, CD95int and CD95hi $T_{SCM}$ cells, respectively; $p_N$, $p_1$, $p_2$ is the proliferation rate of the $T_N$, CD95int and CD95hi populations, respectively; $\Delta_N$ the fraction of $T_N$ undergoing differentiation per day, $k$ the size of the clonal burst, $f$ the fraction of expanded cells that differentiate into the CD95int population, $R_1$ the ratio of CD95int to $T_N$ cells, $R_2$ the ratio of CD95hi to $T_N$ cells and $\delta$ is the number of bases lost per division (assumed to be 50 bp).

**Model D. Linear differentiation model (CD95int first).**

$$\dot{\theta} = \left( \frac{R_1}{R_1 + R_2} \dot{\mu}_1 + \frac{R_2}{R_1 + R_2} \dot{\mu}_2 - \dot{\mu}_N \right) \delta$$
(17)

with

$$\dot{\mu}_1 = 2p_1 + 2^k \Delta_N(\mu_N + k - \mu_1)/R_1$$
$$\dot{\mu}_2 = 2p_2 + \Delta_1(\mu_1 - \mu_2)R_1/R_2$$
$$\dot{\mu}_N = 2p_N$$

Where $\mu_N$, $\mu_1$, $\mu_2$ is the length of telomeres in $T_N$, CD95int and CD95hi $T_{SCM}$ cells, respectively; $p_N$, $p_1$, $p_2$ is the proliferation rate of the $T_N$, CD95int and CD95hi populations, respectively; $\Delta_N$ the fraction of $T_N$ undergoing differentiation per day, $\Delta_1$ the rate of differentiation from CD95int to CD95hi, $k$ the size of the clonal burst, $R_1$ the ratio of CD95int to $T_N$ cells, $R_2$ the ratio of CD95hi to $T_N$ cells, $\delta$ is the number of bases lost per division (assumed to be 50 bp).

**Model E. Linear differentiation model (CD95hi first).**

$$\dot{\theta} = \left( \frac{R_1}{R_1 + R_2} \dot{\mu}_1 + \frac{R_2}{R_1 + R_2} \dot{\mu}_2 - \dot{\mu}_N \right) \delta$$
where
$$\dot{\mu}_2 = 2p_2 + 2^k \Delta_N(\mu_N + k - \mu_2)/R_2$$
$$\dot{\mu}_1 = 2p_1 + \Delta_1(\mu_2 - \mu_1)R_2/R_1$$
$$\dot{\mu}_N = 2p_N$$

(18)

Where $\mu_N$, $\mu_1$, $\mu_2$ is the length of telomeres in $T_N$, CD95int $T_{SCM}$ and CD95hi $T_{SCM}$ cells, respectively; $p_N$, $p_1$, $p_2$ is the proliferation rate of the $T_N$, CD95int and CD95hi populations, respectively; $\Delta_N$ the fraction of $T_N$ undergoing differentiation per day, $\Delta_1$ the rate of differentiation from CD95hi to CD95int, $k$ the size of the clonal burst, $R_1$ the ratio of CD95int to $T_N$ cells, $R_2$ the ratio of CD95hi to $T_N$ cells and $\delta$ is the number of bases lost per division (assumed to be 50 bp).

**Modelling the data: (3) Frequency of YFV-specific T_SCM cells**

The proportion of tetramer⁺ $T_{SCM}$ cells expressed as a fraction of CD8⁺ CD16⁻ lymphocytes as a function of time since vaccination was calculated for all 5 models by assuming there was no further flow from the $T_N$ pool into the $T_{SCM}$ pool ($\Delta_N = 0$).

   **Model A. Independent homogeneous model and Model B. Independent heterogeneous model.** For models A and B, since the populations are independent, there is no flow from the $T_N$ pool and so, since the populations are independently at equilibrium, both models predict that the YFV-specific $T_{SCM}$ population is of constant size with time post-vaccination.

$$Y(t) = 2A$$

(19)

Where $Y(t)$ is the frequency of the YFV-specific $T_{SCM}$ population at time $t$ and $2A$ is the frequency of the YFV-specific $T_{SCM}$ population at time 0.

   **Model C. Forked differentiation model.**

$$Y(t) = T_1(t) + T_2(t)$$
$$T_1(t) = Ae^{-\alpha t}$$
$$T_2(t) = Ae^{-\gamma t}$$
where
$$\alpha = \Delta_N f 2^k / R_1$$
$$\gamma = \Delta_N(1 - f)2^k / R_2$$

(20)

Where $Y(t)$ is the frequency of the YFV-specific T$_{SCM}$ population at time $t$, T$_1$($t$) is the frequency of the YFV-specific CD95int T$_{SCM}$ population at time $t$, T$_2$($t$) is the frequency of YFV-specific CD95hi T$_{SCM}$ population at time $t$ and 2$A$ is the frequency of the YFV-specific T$_{SCM}$ population at time 0. $\Delta_N$ is the fraction of T$_N$ undergoing differentiation per day, $k$ the size of the clonal burst, $f$ the fraction of expanded cells that differentiate into the CD95int population, $R_1$ the ratio of Tint to T$_N$ cells, $R_2$ the ratio of CD95hi to T$_N$ cells.

### Model D. Linear differentiation model (CD95int first).

$$Y(t) = T_1(t) + T_2(t)$$
$$T_1(t) = Ae^{-\alpha t}$$
$$T_2(t) = \frac{(\gamma-\alpha-\Delta_1)Ae^{-\gamma t}}{\gamma-\alpha} + \frac{A\Delta_1 e^{-\alpha t}}{\gamma-\alpha}$$
where
$$\alpha = \Delta_N 2^k / R_1$$
$$\gamma = \Delta_1 R_1 / R_2$$

(21)

Where $Y(t)$ is the frequency of the YFV-specific T$_{SCM}$ population at time $t$, T$_1$($t$) is the frequency of the YFV-specific CD95int T$_{SCM}$ population at time $t$, T$_2$($t$) is the frequency of YFV-specific CD95hi T$_{SCM}$ population at time $t$ and 2$A$ is the frequency of the YFV-specific T$_{SCM}$ population at time 0. $\Delta_N$ the fraction of T$_N$ undergoing differentiation per day, $\Delta_1$ the rate of differentiation from CD95int to CD95hi, $k$ the size of the clonal burst, $R_1$ the ratio of CD95int to T$_N$ cells, $R_2$ the ratio of CD95hi to T$_N$ cells.

### Model E. Linear differentiation model (CD95hi first).

$$Y(t) = T_1(t) + T_2(t)$$
$$T_2(t) = Ae^{-\alpha t}$$
$$T_1(t) = \frac{(\gamma-\alpha-\Delta_1)Ae^{-\gamma t}}{\gamma-\alpha} + \frac{A\Delta_1 e^{-\alpha t}}{\gamma-\alpha}$$
where
$$\alpha = \Delta_N 2^k / R_2$$
$$\gamma = \Delta_1 R_2 / R_1$$

(22)

Where $Y(t)$ is the frequency of the YFV-specific T$_{SCM}$ population at time $t$, T$_1$($t$) is the frequency of the YFV-specific CD95int T$_{SCM}$ population at time $t$, T$_2$($t$) is the frequency of YFV-specific CD95hi T$_{SCM}$ population at time $t$ and 2$A$ is the frequency of the YFV-specific T$_{SCM}$ population at time 0. $\Delta_N$ the fraction of T$_N$ undergoing differentiation per day, $\Delta_1$ the rate of differentiation from CD95hi to CD95int, $k$ the size of the clonal burst, $R_1$ the ratio of Tint to T$_N$ cells, $R_2$ the ratio of CD95hi to T$_N$ cells.

## Fitting the data

A multistep fitting procedure was used. First the saliva data was fitted to estimate $f$ and $\delta$ for each individual. Then these values were used to estimate $b_w$ and $r_2$ for each individual. And then finally this value of $b_w$ was used in the equations to fit the labelling data. For this final step two separate fits were performed. In one all data (labelling, telomere, YF-vaccination) were fitted simultaneously and in the second only the labelling data were fitted. When fitting, fits were performed separately for the three individuals with labelling data (DW19, DW20, DW25). The YF vaccination data was cross-sectional data from 37 individuals (none of whom were labelled). Therefore, for each of the three labelled individuals we aimed to predict the rate of decline of YFV-specific T$_{SCM}$ based on the individual's kinetic parameters and this was compared with the observed rate of YFV-specific T$_{SCM}$ decline in the cross-sectional cohort. Similarly, the individuals who received the stable isotope label were different from the individuals whose telomere lengths were measured. We therefore used the parameters from the individuals who were labelled to predict the average value of $\theta$, this was compared with the five observed values of $\theta$. For Models A and B, $\theta$ was dependent on age so for these models we predicted $\theta$ for an individual of age 34 yrs (the median age of the

individuals providing telomere data). For Models C–E, $\theta$ is independent of age as these systems reach equilibrium. We therefore compared equilibrium $\theta$ (calculated numerically) with the observed $\theta$. For each type of data (labelling, telomere, YFV) the residuals (difference between the observation and prediction) were normalized by the mean of the data to ensure that the residuals arising from the different types of data were on a similar scale.

Models were fitted by minimizing the sum of squared (normalized) residuals using the global optimizer pseudo from the package FME [64] (v1.3.6.2) in R v 4.2.3. Other packages used in the analysis included deSolve (v1.34), debug (v1.3.189) and pryr (v0.1.6).

## Calculations of clonal half-life

We wanted to estimate the half-life of a T cell clone as this is related to the half-life of immune memory, rather than the more typically estimated population half-life which is related to the half-life of cells.

For Model C (fork) the clonal half-lives are given by

$$
\begin{aligned}
\text{clonal half} - \text{life CD95int} &= \frac{\ln(2)}{d_1 - p_1} \\
&= \frac{R_1 \ln(2)}{\Delta_N f 2^k} \\
\text{clonal half} - \text{life CD95hi} &= \frac{\ln(2)}{d_2 - p_2} \\
&= \frac{R_2 \ln(2)}{\Delta_N (1-f) 2^k}
\end{aligned}
\tag{23}
$$

For Model D (linear CD95int first) the clonal half-lives are given by

$$
\begin{aligned}
\text{clonal half} - \text{life CD95int} &= \frac{\ln(2)}{d_1 + \Delta_1 - p_1} \\
&= \frac{R_1 \ln(2)}{\Delta_N 2^k} \\
\text{clonal half} - \text{life CD95hi} &= \frac{\ln(2)}{d_2 - p_2} \\
&= \frac{R_2 \ln(2)}{R_1 \Delta_1}
\end{aligned}
\tag{24}
$$

For Model E (linear CD95hi first) the clonal half-lives are given by

$$
\begin{aligned}
\text{clonal half} - \text{life CD95int} &= \frac{\ln(2)}{d_1 - p_1} \\
&= \frac{R_1 \ln(2)}{R_2 \Delta_1} \\
\text{clonal half} - \text{life CD95hi} &= \frac{\ln(2)}{d_2 + \Delta_1 - p_2} \\
&= \frac{R_2 \ln(2)}{\Delta_N 2^k}
\end{aligned}
\tag{25}
$$

In order to calculate clonal half-lives and the uncertainty in the clonal half-lives for each individual the data were randomly sampled with replacement to produce a dataset with the same number of points, the model was fit to the data, the best fit parameters estimated (using 30 different random seeds to find the global minima) and the corresponding clonal half-lives calculated. This process was repeated 100 times. The median and interquartile ranges of the parameters and half-lives are reported. We calculate the clonal half-lives from the bootstrap estimates rather than by using the point estimates of the parameters due to potential correlation between the parameters.

## Calculations of residency time

We report the residency time (average time that a cell remains in a given population) for CD95int TSCM and CD95hi TSCM calculated using Model D (linear Tint first).

$$residencytimeCD95int = \frac{1}{d_1 + \Delta_1}$$

$$residencytimeCD95hi = \frac{1}{d_2}$$

## Model predictions

For each of the 5 models considered, the best fit parameters (Table 1 and Tables F–I in S2 Text) and corresponding model equations were used to predict the dynamics of virus-specific CD95int $T_{SCM}$ and CD95hi $T_{SCM}$ over time for 35 yrs. These predictions were compared (simply by overlaying, not fitting) with data from Fuertes Marraco and colleagues [2] in which they quantified "Naïve-hi" (CD8⁺CCR7⁺⁺CD45RA⁺⁺) $T_{SCM}$ cells specific for the HLA-A*02-restricted YFV NS4b214-222 epitope which are enriched for CD95int $T_{SCM}$ and the remaining "naïve-int" tetramer-positive $T_{SCM}$ which will be enriched for CD95hi $T_{SCM}$. In both cases the tetramer-positive cells were expressed as a percentage of the total CD8⁺CD3⁺ lymphocyte population. The initial condition for CD95int and CD95hi $T_{SCM}$ populations was taken to be the median of CD95int and the median of CD95hi $T_{SCM}$ sampled within 1 yr of vaccination, respectively.

## Model independent analysis of cross-sectional YFV-tetramer data

We follow the approach of Fuertes Marraco and colleagues [2] who quantified the frequency of YFV-specific cells within each subpopulation rather than within the total CD8⁺ population (e.g., quantify YFV-specific $T_{CM}$ CD8⁺ cells as a percent of total $T_{CM}$ rather than as a percent of total CD8⁺ T cells) in order to reduce between-individual variation in the relative sizes of different CD8⁺ subpopulations. She then expressed the frequency $log_{10}$ and estimated the rate of decline of the YFV-specific subpopulations by linear regression. She also made two further changes: excluding three individuals with very low frequencies of YFV-specific CD8⁺ T cells and including 4 individuals who had received multiple YF vaccinations. We report results from both approaches.

## Statistics

All statistics were calculated in R v4.2.3. All *P* values reported are two tailed and where multiple tests were conducted the number of independent tests is noted. In the case of box plots, the bar represents the median, the box the interquartile range (IQR) and whiskers show lower quartile − 1.5 * IQR and upper quartile + 1.5 * IQR; all data (including outliers) are shown.

## Supporting information

**S1 Text. Supplementary Figures. Fig A.** Ki67 varies significantly in an independent dataset **Fig B.** Representative Gating. **Fig C.** Phenotype of sorted cells following 7 days *ex vivo* culture. **Fig D.** CTV profile of sorted cells following 7 days *ex vivo* culture. **Fig E.** CD95hi TSCM cells exhibit increased functionality compared to both CD95int $T_{SCM}$ cells and $T_{CM}$ cells. **Fig F.** Phenotypic characterization of CD8⁺ memory subsets. **Fig G.** Ki67 expression after exclusion of PD1⁺TIGIT⁺ cells. **Fig H.** Heat maps of the 70 most differentially expressed genes. **Fig I.** CD95hi cells are transcriptionally and translationally closer to $T_{CM}$ while CD95int more closely resemble TN. **Fig J.** Schematic of Mechanistic Model to Describe Monocyte Kinetics. **Fig K.** Best fits of the models to the saliva and monocyte data. (A). **Fig L.** Fit of Model A (independent homogeneous populations) to all data (labelling, telomere, YFV) simultaneously. **Fig M.** Fit of Model B (independent heterogeneous populations) to all data (labelling, telomere, YFV) simultaneously. **Fig N.** Fit of Model C (fork) to all data (labelling, telomere, YFV) simultaneously. **Fig O.** Fit of Model E (linear, CD95hi first) to all data (labelling, telomere, YFV) simultaneously. **Fig P.** CD45A and CCR7 expression of the CD95hi $T_{SCM}$ and CD95int $T_{SCM}$ population. **Fig Q.** Predictions of Model C (fork) overlaid (NOT fitted) onto a

novel data set. **Fig R.** Predictions of Model E (linear Thi first) overlaid (NOT fitted) onto a novel data set. **Fig S.** Duplicate of Fig 2 showing unique identifier for each donor in response to request from reviewer.
(PDF)

**S2 Text.  Supplementary Tables. Table A.** Antibodies used for flow cytometry and sorting. **Table B**. Human TaqMan probes used for RT-PCR. **Table C**. RNAseq Experimental Design. **Table D.** Saliva and monocyte parameters. **Table E.** Parameters in the models for fitting labelling, telomere and YF-vaccination data. **Table F.** Parameters obtained by fitting model A (indep homog) to labelling, telomere and YFV data. **Table G.** Parameters obtained by fitting model B (indep heterog) to labelling, telomere and YFV data. **Table H.** Parameters obtained by fitting model C (fork) to labelling, telomere and YFV data. **Table I.** Parameters obtained by fitting model E (linear, CD95hi first) to labelling, telomere and YFV data. **Table J.** Summary of the fits of the 5 models to the labelling, telomere and YFV vaccination data. **Table K.** Summary of the fits of the 5 models to the labelling data only. **Table L.** Summary of the comparison of the 5 model predictions with the unseen, unfitted YFV vaccination data.
(PDF)

**S1 Data.  Underlying data.**
(XLSX)

**S1 File.  Flow data underlying supplementary figures (PART I).**
(ZIP)

**S2 File.  Flow data underlying supplementary figures (PART II).**
(ZIP)

**S3 File.  Flow data underlying Fig 2 (main text).**
(ZIP)

## Acknowledgments

We are very grateful to Silvia Fuertes Marraco for sharing her published data.

## Author contributions

**Conceptualization:** Danai Koftori, Laura Mora Bitria, Becca Asquith.

**Data curation:** Danai Koftori, Charandeep Kaur, Yan Zhang, Piotr F Burzyński, Kristin Ladell, Katrina M Pollock, Derek Macallan, Becca Asquith.

**Formal analysis:** Danai Koftori, Laura Mora Bitria, Yan Zhang, Ada W C Yan, Piotr F Burzyński, Kristin Ladell, Katrina M Pollock, Derek Macallan, Becca Asquith.

**Funding acquisition:** Becca Asquith.

**Investigation:** Danai Koftori, Charandeep Kaur, Yan Zhang, Linda Hadcocks, Ada W C Yan, Piotr F Burzyński, Kristin Ladell, Daniel E Speiser, Katrina M Pollock, Derek Macallan, Becca Asquith.

**Methodology:** Danai Koftori, Laura Mora Bitria, Becca Asquith.

**Project administration:** Danai Koftori, Yan Zhang, Linda Hadcocks, Becca Asquith.

**Resources:** Becca Asquith.

**Supervision:** Daniel E Speiser, Katrina M Pollock, Derek Macallan, Becca Asquith.

**Validation:** Danai Koftori, Laura Mora Bitria, Yan Zhang, Ada W C Yan, Derek Macallan, Becca Asquith.

**Visualization:** Laura Mora Bitria, Becca Asquith.

**Writing – original draft:** Becca Asquith.

**Writing – review & editing:** Danai Koftori, Charandeep Kaur, Laura Mora Bitria, Daniel E Speiser, Derek Macallan, Becca Asquith.

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
