## [Editor Report · Decision Letter 0]

Dear Dr Asquith,

Thank you for submitting your manuscript entitled "Two distinct subpopulations of human stem-like memory T cells with different dynamics, function and homing properties" for consideration as a Research Article by PLOS Biology.

Your manuscript has now been evaluated by the PLOS Biology editorial staff, as well as by an academic editor with relevant expertise, and I am writing to let you know that we would like to send your submission out for external peer review.

Once your full submission is complete, your paper will undergo a series of checks in preparation for peer review. After your manuscript has passed the checks it will be sent out for review. To provide the metadata for your submission, please Login to Editorial Manager (https://www.editorialmanager.com/pbiology) within two working days, i.e. by Aug 09 2024 11:59PM.

Kind regards,

Melissa

Melissa Vazquez Hernandez, Ph.D.

Associate Editor

PLOS Biology

---

## [Decision Letter · Decision Letter 1]

Dear Dr Asquith,

Thank you for your patience while your manuscript "Two distinct subpopulations of human stem-like memory T cells with different dynamics, function and homing properties" was peer-reviewed at PLOS Biology. It has now been evaluated by the PLOS Biology editors, an Academic Editor with relevant expertise, and by several independent reviewers.

In light of the reviews, which you will find at the end of this email, we would like to invite you to revise the work to thoroughly address the reviewers' reports. The reviewers are positive about the relevance of the findings but have raised several concerns that need to be addressed before we can consider the study further. Reviewer 1 requests the quantification of the two populations over an individual's lifetime and suggesting speculation on their kinetic behaviour in vivo. Reviewer 2 would like some clarifications regarding the claim that both populations are stem-like, potential drivers of conversion between cells, the relationship between TSCM and memory subsets, and mean residence time. The reviewer also suggests ways to strengthen the statistical power of the mathematical model. Reviewer 3 would like you to compare the data with Zhao et al. 2021 (10.1002/eji.202049057), as well as the suggestions of include a CellTraceViolet flow histogram, live/dead staining in the gating strategy, and heatmaps of the most variable genes.

Given the extent of revision needed, we cannot make a decision about publication until we have seen the revised manuscript and your response to the reviewers' comments. Your revised manuscript is likely to be sent for further evaluation by all or a subset of the reviewers.

**IMPORTANT - SUBMITTING YOUR REVISION**

*Re-submission Checklist*

*Published Peer Review*

*PLOS Data Policy*

*Blot and Gel Data Policy*

Sincerely,

Melissa

Melissa Vazquez Hernandez, Ph.D.

Associate Editor

PLOS Biology

REVIEWERS' COMMENTS:

Reviewer #1:

This is a very thoughtful and interesting manuscript which uses flow cytometric, cell labelling, and molecular techniques to distinguish the activity of two human T memory stem cell subpopulations. The authors then generate and test five mathematical models linking the two subsets, and retest their analyses using a completely different set of cell surface markers on a different clinical T cell population from a published dataset. Overall, the work is very thoughtful and well-executed. In fact, I only have one experimental and two textual, requests for revision. First, given their relative numbers over time, can the authors quantitate the approximate numbers of the two populations (in LN + Blood) over an individual's lifetime; 2) I would invite the authors to speculate on how the intermediate and high populations function kinetically in vivo: i.e. does the intermediate population divide much more infrequently than the high population; (this would be similar to the recent claims about slightly more mature hematopoietic cells (MPPs) than HSCs demonstrating both self-renewal and ebullient proliferative differentiation; and 3) I do think the authors should note that the first suggestion of T Memory Stem Cells in mice was published by Zhang, Emerson et al. in 2005 in Nature Medicine (in mice, Sca1 was a useful marker), just to be fair and accurate.

Reviewer #2:

This is an extensive study that makes the case for two distinct lineages within human stem-like memory T cells, distinguished by their levels of CD95 expression. The authors previously inferred heterogeneity within TSCM (del Amo Plos Biol 2018) but did not identify a marker differentiating the two subsets. This latest study combines an even wider array of assays and mathematical modelling tools - the force of the combined approach is impressive. It is established that CD95int cells divide and are lost at slow and almost-balanced rates, giving very long clonal lifespans. CD95hi cells are more proliferative but also die rapidly, giving shorter clonal lifespans. The relationship between the lineages is not definitively established but the data are consistent with a linear model in which the slow CD95int cells continually seed faster and shorter-lived CD95hi cells.

it wasn't totally clear to me why the two populations are collectively stem-like (abstract). If one thinks of clonal persistence it seems that the CD95int cells really are the stem-like candidate.

It would be good to discuss further the potential drivers of int->hi conversion. It is modelled as constitutive but could this be an intermittent, antigen-driven process? If it's recruitment from naive cells does this mean, in the case of YFV, that

The 7 day assays looking at outcomes of IL15 and IL2 stimulation are really interesting in this regard. Could you use that information to help speculate how these TSCM subsets are related and embedded within memory subsets as a whole? They are in vitro assays but they do have some directionality information in there.

it would be good to quote cell lifespans as well as clonal half-lives. Alternatively, you could combine death and differentiation to define a mean residence time 1/(delta + Delta). I think timescales are a bit easier for everyone to digest than rate constants. A related parameter is then Delta/(Delta + d) which is the proportion cells lost from CD95int population that seeds the CD95hi subset - a sort of conversion efficiency. Maybe a figure with all of these relationships and rates?

The topology question is very interesting. The authors use goodness-of-fit criteria to establish support. As is also often the case, the fits of the different models are visually quite similar so we are putting a lot of trust in the statistics. The AIC values presumably include residuals from multiple datasets - are these weighted in any way? It's usual to normalise each set of squared residuals by the variance of the observations in that dataset - those variances are actually free parameters, though not usually estimated with simple least squares performed with one dataset, where you can use the AICc formula with the SSR directly. It becomes more important when combining multiple datasets with different levels of uncertainty in the observations. With least squares fitting you can skirt around this by plugging in the ML estimate of each dataset's variance (see Hogan PNAS 2015) - this essentially gives you a set of weights for the SSRs from different datasets (there are other more rigorous ways to do it in Bayesian frameworks).

I don't want to labour this point as it's a bit technical but for me the relationship between these two subsets is the most intriguing aspect of the paper and so it would be great to see more of the methodology.

Reviewer #3:

Background: The work by Koftori, Asquith and colleagues builds upon seminal work by Gattinoni et al 2011 who identified a novel T cell subset with stem cell like properties, defined as naïve phenotype and CD95+. This stem cell memory populated had the ability to self-renew and repopulate the memory subsets. An important follow-up study by Costa del Amo et al 2018 studied the dynamics of stem cell memory revealing that stem cell memory are likely composed of two subsets. One subset has properties of a long-lived cell while the other is short lived.

Current work: In the current study Koftori demonstrates the stem cell memory can be divided into two subpopulations: those that express high levels of CD95 (CD95hi) and those that express intermediate levels of CD95 (CD95int). They show these two populations are distinct in many ways including (1) their half-lives and (2) their ability to self-renew. Specifically, the CD95hi cells had a much shorter half-life and a greater ability to self-renew and were more similar to CM while the CD95int shared characteristics with naïve T cells. Finally the authors utilize a modeling approach to suggest that naïve T cells give rise to CD95int which give rise to CD95hi which give rise to CM. This model builds on several sources of data and is consistent with the data in the figures as well. Moreover, the model invites new avenues of research.

Context: Notably, other group have subdivided stem cell memory into subpopulations based on different surface markers. In particular, Zhao et al 2021 subdivided TSCM into CD122 hi and CD122 low. However, this prior study using CD122 was less convincing. Specifically, the sorted populations did not convincingly separate the ability to self-renew; nor did they even measure the half-life. This is an important omission since it is the entire basis for the hypothesis that there were two subsets within the stem cell memory population.

Data and analysis: Almost all of the data that would be needed to replicate the study are available, except for Figures 2-4, comment 2-4. The statistical analysis is rigorous and robust.

Major Comments:

1. It appears the authors have been able to separate the frequently dividing cells from the slow dividing subset better than prior studies. However, the data should be more comparable to the prior study by Zhao et al. 2021 so that the reader would be more informed when choosing markers to sort separate the stem cell memory population. In particular, the authors should sort 4 populations (CD95hi+CD122hi; CD95hi+CD122lo; CD95int+CD122hi; CD95int+CD122lo and repeat the experiments in Figure 2A and 2C.)

2. Figure 2: The authors should show the flow histogram analysis of their CellTraceViolet along with the proliferation index (akin to Zhao et al Eur. J. Immunol 2021) for CD95 hi and CD95int. Again this would make their study more comparable to the Zhao study.

3. Supplementary figure 2 provides gating strategy. This gating strategy does not include the live/dead staining nor the cell trace staining. The authors should provide the full gating strategy starting with FSC vs SSC.

4. Please provide heat maps of the most variable genes (akin to Zhao 2021) to show the differentially expressed genes in the different subsets with and without stimulation. It is more intuitive to see similarities between subsets with the heat map.

5. Figure 3 A and B(qPCR) comparing fold change gene expression in CD95int/CD95hi) is not very convincing to me. The authors could delete it. The data tables while transparent leave too much work for the reader/reviewer. If the authors prefer to keep this data, they should limit the analysis to genes that were actually differentially expressed between the critical subsets in their data set and remove gene sets that were not detected in any samples.

Minor Comments:

1. Figure 2 would benefit from unique identifiers.

2. The sentence "The only T cell pathway (CSE21360, PRIMARY vs QUATERNARY MEMORY CD8 T CELL UP from 19) implies that genes upregulated in CD95hi Tscm are enriched in genes upregulated in memory CD8 T cells following multiple antigen challenge whereas genes upregulated in CD95int are associated with memory CD8 T cells following primary antigen challenge" should be further clarified.

3. Please fix the y axis label in Figure 7b

4. The telomere analysis from a published dataset could be deleted as it is not needed to make the conclusions of the story.

---

## [Decision Letter · Decision Letter 2]

Dear Dr Asquith,

Thank you for your patience while we considered your revised manuscript "Two distinct subpopulations of human stem-like memory T cells with different dynamics, function and homing properties" for publication as a Research Article at PLOS Biology. This revised version of your manuscript has been evaluated by the PLOS Biology editors, the Academic Editor and one of the original reviewers.

I would like to truly apologize for the incredible long time it took to re-review. Unfortunately, one reviewer declined to re-review and we were not able to contact another reviewer that had agreed to re-review, which delayed the process.

Based on the review of Reviewer 3 and on our Academic Editor's assessment of your revision, we are likely to accept this manuscript for publication, provided you satisfactorily address the remaining points raised by the reviewer and the editorial requests. Please also make sure to address the following data and other policy-related requests.

a) We routinely suggest changes to titles to ensure maximum accessibility for a broad, non-specialist readership, and to ensure they reflect the contents of the paper. In this case, we would suggest a minor edit to the title, as follows. Please ensure you change both the manuscript file and the online submission system, as they need to match for final acceptance:

"Two distinct subpopulations of human stem-like memory T cells exhibit complementary roles in self-renewal and clonal longevity"

b) The Ethics statement needs to be the first subheading in the Methods sections the Material & Methods section. You currently have it under “Study subjects”; please move it before this section.

Please supply the numerical values either in the a supplementary file or as a permanent DOI’d deposition for the following figures:

Figure 1AB, 2A-E, 4AB, 5AB, 6B, 8ABC, 9AB, 10AB, 12, S1, S3AB, S4A-D, S5AB, S6AB, S7AB, S8AB, S9ABC, S11AB, S12ABC, S13ABC, S14ABC, S16ABC, S17, S18, and Table 1

d) Please cite the location of the data clearly in all relevant main and supplementary Figure legends, e.g. “The data underlying this Figure can be found in S1 Data” or “The data underlying this Figure can be found in https://doi.org/10.5281/zenodo.XXXXX”

e) For figures containing FACS data (Figures S2ABC, S16), please provide the FCS files and a picture showing the successive plots and gates that were applied to the FCS files to generate the figure. We ask that you please deposit this data in the FlowRepository (https://flowrepository.org/) and provide the accession number/URL of the deposition in the Data Availability Statement in the online submission form.

f) Please ensure that your Data Statement in the submission system accurately describes where your data can be found and is in final format, as it will be published as written there.

g) Per journal policy, if you have generated any custom code during the course of this investigation, please make it available without restrictions upon publication. Please ensure that the code is sufficiently well documented and reusable, and that your Data Statement in the Editorial Manager submission system accurately describes where your code can be found.

We expect to receive your revised manuscript within two weeks.

*Published Peer Review History*

*Press*

Sincerely,

Melissa

Melissa Vazquez Hernandez, Ph.D.

Associate Editor

PLOS Biology

REVIEWER'S COMMENTS:

Reviewer #3:

We thank the authors for their thoughtful responses to our comments and would like to accept the manuscript. Overall, it is an important contribution and a large body of work and the reviewers were responsive to all of my queries.

It is unfortunate that sorting the cells into CD95hi/int CD122hi/int yielded low cell numbers, but we appreciate the effort.

One minor note to discuss is the authors mention in their abstract and manuscript that the CD95hi cells have considerably higher proliferation than the CD95int cells. The in vivo data in Figure 6 supports their claim, however there is inter-donor variability. The authors could discuss the variability in their data which could easily related to variable after cell sorting.

I appreciate the addition of the RNA heat maps in Supplemental Figure 8. Could the authors move LD2_int_U with the other intermediate cells in Supplemental Figure 8a.

I prefer the unique identifiers in figure 2 as it helps the reader understand donor variability.

---

## [Editor Report · Decision Letter 3]

Dear Becca,

Thank you for the submission of your revised Research Article "Two distinct subpopulations of human stem-like memory T cells exhibit complementary roles in self-renewal and clonal longevity" for publication in PLOS Biology. On behalf of my colleagues and the Academic Editor, Avinash Bhandoola, I am pleased to say that we can in principle accept your manuscript for publication, provided you address any remaining formatting and reporting issues. These will be detailed in an email you should receive within 2-3 business days from our colleagues in the journal operations team; no action is required from you until then. Please note that we will not be able to formally accept your manuscript and schedule it for publication until you have completed any requested changes.

PRESS

Sincerely, 

Melissa

Melissa Vazquez Hernandez, Ph.D., Ph.D.

Associate Editor

PLOS Biology
